# Cortical astrocyte N-methyl-D-aspartate receptors influence whisker barrel activity and sensory discrimination in mice

Noushin Ahmadpour[1,6], Meher Kantroo [1,6], Michael J. Stobart[1], Jessica Meza-Resillas [1], Shahin Shabanipour[1], Jesus Parra-Nuñez[1], Tetiana Salamovska[1], Anna Muzaleva[1], Finnegan O'Hara [1], Dustin Erickson [1], Bruno Di Gaetano[1], Sofia Carrion-Falgarona[1], Bruno Weber [2], Alana Lamont [3,4], Natalie E. Lavine[3,4], Tiina M. Kauppinen [3,4], Michael F. Jackson [3,4] & Jillian L. Stobart [1,5] ✉

Astrocytes express ionotropic receptors, including N-methyl-D-aspartate receptors (NMDARs). However, the contribution of NMDARs to astrocyte-neuron interactions, particularly in vivo, has not been elucidated. Here we show that a knockdown approach to selectively reduce NMDARs in mouse cortical astrocytes decreases astrocyte $Ca^{2+}$ transients evoked by sensory stimulation. Astrocyte NMDAR knockdown also impairs nearby neuronal circuits by elevating spontaneous neuron activity and limiting neuronal recruitment, synchronization, and adaptation during sensory stimulation. Furthermore, this compromises the optimal processing of sensory information since the sensory acuity of the mice is reduced during a whisker-dependent tactile discrimination task. Lastly, we rescue the effects of astrocyte NMDAR knockdown on neurons and improve the tactile acuity of the animal by supplying exogenous ATP. Overall, our findings show that astrocytes can respond to nearby neuronal activity via their NMDAR, and that these receptors are an important component for purinergic signaling that regulate astrocyte-neuron interactions and cortical sensory discrimination in vivo.

Astrocytes are glial cells that are essential for homeostasis and have increasingly been shown to influence animal behavior[1–4]. Morphologically, astrocytes integrate into circuits through their highly branched processes that enwrap nearby synapses. Here, astrocytes can modulate neuronal activity through the $Ca^{2+}$-mediated release of signaling molecules (i.e., gliotransmitters[5,6]). Since astrocytes form an interconnected glial network and a single astrocyte contacts multiple synapses[5], local activity sensed by an astrocyte is potentially relayed to influence neurons at remote sites and at the population level[5,7].

However, the conditions and pathways involved in the astrocytic modulation of neuronal populations and animal behavior are not clearly understood, particularly within the cortex.

An essential component of the bidirectional crosstalk between astrocytes and neurons is astrocytic calcium signaling[6]. Astrocyte $Ca^{2+}$ dynamics primarily occur as $Ca^{2+}$ microdomains, where intracellular $Ca^{2+}$ is elevated within localized subcellular compartments located in the soma, endfeet, or fine processes[8–12]. Following nearby synaptic activity, such as that induced by somatosensory whisker stimulation,

[1]College of Pharmacy, University of Manitoba, Winnipeg, MB, Canada. [2]Institute of Pharmacology and Toxicology, University of Zurich, Zurich, Switzerland. [3]Department of Pharmacology and Therapeutics, University of Manitoba, Winnipeg, MB, Canada. [4]PrairieNeuro Research Center, Health Sciences Center, Winnipeg, MB, Canada. [5]Centre on Aging, University of Manitoba, Winnipeg, MB, Canada. [6]These authors contributed equally: Noushin Ahmadpour, Meher Kantroo. ✉e-mail: jillian.stobart@umanitoba.ca

there is an increase in the number of astrocyte $Ca^{2+}$ microdomains, as well as their size ($\mu m^2$) and signal amplitude[8,9,11,13]. The majority of astrocyte $Ca^{2+}$ events evoked by local circuit activity have a delayed signal onset relative to the dynamics of nearby neurons[8,14]. However, fast-onset astrocyte $Ca^{2+}$ microdomains have been reported, which occur on the spatial scale of neuronal spines and closely parallel the time course of neuronal $Ca^{2+}$ events[8,13]. Therefore, astrocyte $Ca^{2+}$ signaling has the necessary temporal and spatial properties for rapid synaptic modulation.

A growing body of evidence suggests that ion channels and ionotropic receptors can evoke astrocyte $Ca^{2+}$ microdomains via transmembrane $Ca^{2+}$ flux[12,15]. Astrocytes express ionotropic N-methyl-D-aspartate receptors (NMDARs), which are activated by synaptically released glutamate in the presence of D-serine or glycine. Studies of astrocyte NMDARs have focussed on in vitro preparations or brain slices[16–26], and indicate that these receptors are primarily composed of GluN1 as the essential subunit, paired with GluN2C, 2D or GluN3 subunits[23,26,27] that are not sensitive to blockade by $Mg^{2+}$ ions[25,26]. When activated, these receptors induce astrocyte membrane depolarizations, inward currents, and increases in intracellular $Ca^{2+}$[18,23,25,26,28]. NMDAR pharmacological approaches have suggested that these receptors do not mediate resting astrocyte $Ca^{2+}$ microdomains in the hippocampus[29], but contribute to stimulation-evoked astrocyte $Ca^{2+}$ increases in multiple brain regions[17,30,31] and perivascular endfeet[14,32]. However, all of these studies used drugs that also inhibited neuronal NMDARs, confounding the interpretation of these results.

In this study, we aimed to determine the contribution of astrocyte NMDARs to $Ca^{2+}$ microdomains and their impact on nearby neuronal activity during whisker stimulation in awake mice in vivo. We implemented a miRNA-adapted shRNA (shRNA$^{mir}$) strategy to knockdown (KD) the gene for the essential GLUN1 NMDAR subunit, Grin1, selectively in cortical astrocytes of the barrel cortex (i.e., reduced Grin1 expression would eliminate all astrocyte NMDARs irrespective of their subunit composition). We report a mechanism where astrocyte NMDARs regulate astrocyte $Ca^{2+}$ microdomains evoked by whisker stimulation and play a key role in regulating optimal neuronal activity needed for information processing that dictates sensory discrimination.

## Results

### Knockdown of astrocyte NMDAR expression

To reduce the expression of NMDARs in cortical astrocytes, we used an adeno-associated virus (AAV9) containing the astrocyte-specific gfaABC1D promoter, three unique miRNA-adapted shRNA hairpins for knockdown of Grin1 mRNA (referred to as Grin1 KD), and a membrane-tagged genetically-encoded calcium indicator, Lck-GCaMP6f (AAV9-sGFAP-Grin1-shRNA$^{mir}$-Lck-GCaMP6f; Fig. 1A). A comparable AAV9 containing three non-silencing shRNA$^{mir}$ was used as a control (AAV9-sGFAP-NS-shRNA$^{mir}$-Lck-GCaMP6f; Fig. 1A). Similar viral constructs with this promoter have been widely adopted to target astrocytes[4,8,10,33]. Injection of these AAVs into the whisker barrel somatosensory cortex of adult C57BL/6NCrl mice (Fig. 1B) resulted in astrocyte-specific expression of Lck-GCaMP6f (Fig. 1C, D) in more than 95% of astrocytes in the viral transduction area (Supplementary Fig. 1A–D), without notable increases in GFAP labeling indicative of astrogliosis[11]. When we dissociated cortical tissue and collected astrocytes by Fluorescence-Activated Cell Sorting (FACS) using their GCaMP6f fluorescence (Supplementary Fig. 1E), we found that the expression of Grin1 was reduced at the mRNA level in the Grin1 KD astrocytes by 70% using qPCR (Fig. 1E). RNA-seq of these samples confirmed an enrichment of astrocyte-selective genes (Supplementary Fig. 1F). In considering possible compensation by other NMDAR subunits, we found that Grin1 and Grin2a expression were reduced in Grin1 KD samples, but genes for other subunits were not differentially expressed (Fig. 1F, G). We also transduced astrocyte-enriched glial

cultures with our AAVs and found a reduction in GLUN1 fluorescence by immunocytochemistry in Grin1 KD cells (Fig. 1H–J). Selectively detecting astrocyte GLUN1 in situ by immunohistochemistry proved to be a challenge due to signal contamination from strong GLUN1 expression by nearby neurons. Therefore, we depleted neuronal GLUN1 expression by mixing our astrocyte Grin1 KD or non-silencing control viruses with AAV9-hSYN-mCherry-Cre and injecting the viruses into mice that express floxed Grin1 (Grin1$^{fl/fl}$)[34] and Cre-dependent EYFP (EYFP$^{fl/fl}$)[35]. Decreased GLUN1 fluorescence in the neuropil of the virus injection area was apparent when comparing the fluorescence of the fields of view (FOVs) in the astrocyte non-silencing control virus area vs. FOVs outside the virus area (Supplementary Fig. 2; neuronal GLUN1 was depleted as a result of SYN-driven neuronal Cre). Decreasing neuronal GLUN1 expression made it possible to determine the difference in GLUN1 expression between astrocytes expressing our non-silencing control and Grin1 KD viruses (using GFAP signal as a mask; Fig. 1K, L). GLUN1 fluorescence was reduced by ~75% in astrocytes in the Grin1 KD area (Fig. 1M).

Grin1 KD also resulted in a functional reduction in astrocyte NMDAR activity evoked by agonist application to acute adult brain slices (Fig. 2A). Due to the membrane localization of the Lck-GCaMP6f sensor, localized $Ca^{2+}$ signals in active regions of interest (ROIs; i.e., $Ca^{2+}$ microdomains) were identified throughout the "astropil" using our MATLAB toolbox, CHIPS[8,36,37] (Fig. 2B, C). Following treatment with neuronal blockers (1 $\mu M$ TTX, 10 $\mu M$ CNQX, 100 $\mu M$ CdCl$_2$) and bath application of NMDAR agonists, NMDA (50 $\mu M$) and D-serine (10 $\mu M$), ROIs in control astrocytes responded with a $Ca^{2+}$ increase (Fig. 2B, D). The magnitude of the response was attenuated in Grin1 KD astrocytes (Fig. 2C, D) in terms of the area under the curve (mean ± SEM; 1225.12 ± 167.28 control vs. 134.97 ± 21.15 KD; Fig. 2E) and the amplitude of $Ca^{2+}$ peaks in each ROI (5.40 ± 0.48 dF/F control vs 1.48 ± 0.09 KD during agonists; Fig. 2F). In contrast, astrocyte responses to alternate stimuli that induce intracellular $Ca^{2+}$ elevation were not affected, since both control and Grin1 KD cells responded with similar $Ca^{2+}$ amplitudes to alpha-1 adrenergic receptor agonist, phenylephrine, which induces IP$_3$-mediated release of $Ca^{2+}$ from intracellular stores (2.71 ± 0.21 dF/F control vs. 3.28 ± 0.27 Grin1 KD; Fig. 2G).

### Astrocyte NMDAR knockdown reduces astrocyte $Ca^{2+}$ events in vivo

To investigate the impact of Grin1 KD on $Ca^{2+}$ events in cortical astrocytes in vivo, we prepared a chronic cranial window over the whisker barrel cortex in adult mice injected with AAVs encoding astrocyte-specific Grin1 knockdown or control shRNA$^{mir}$ (with GFAP-Lck-GCaMP6f; Fig. 3A). Following recovery from surgery and mapping of the barrel cortex with intrinsic optical imaging (Fig. 3B), mice were trained for awake two-photon imaging[8] to capture cortical Layer 2/3 (L2/3) astrocyte $Ca^{2+}$ signaling during individual whisker vibrations (90 Hz, 8 s) corresponding to the virus injection area (Fig. 3A). Similar to $Ca^{2+}$ imaging in brain slices, our analysis detected two-dimensional $Ca^{2+}$ microdomains, which were spatially-confined ROIs with active $Ca^{2+}$ events during the imaging trial. Before some imaging sessions, we injected SR101 (20 mg/kg i.v.) which crossed the blood-brain-barrier to label cortical astrocytes within 90 min[38]. This allowed us to manually identify astrocyte somata and endfeet within the "cloud" of Lck-GCaMP6f fluorescence (Fig. 3C, D). The area of each field of view (FOV) was normalized to the total whisker barrel area and the number of ROIs with at least one $Ca^{2+}$ event was represented as the ROIs/barrel area. Multiple trials with and without whisker stimulation were recorded for each FOV (presented as activity maps in Fig. 3E, F).

When considering all ROIs with $Ca^{2+}$ activity, both control and Grin1 KD astrocytes had a similar number of spontaneous microdomains in trials without whisker stimulation (mean ± SEM; 5.33 ± 0.45 control vs. 5.61 ± 0.53 KD ROIs/barrel area; Fig. 3E vs. Fig. 3F; Fig. 3G). As shown previously[8,11,13], whisker stimulation evoked more

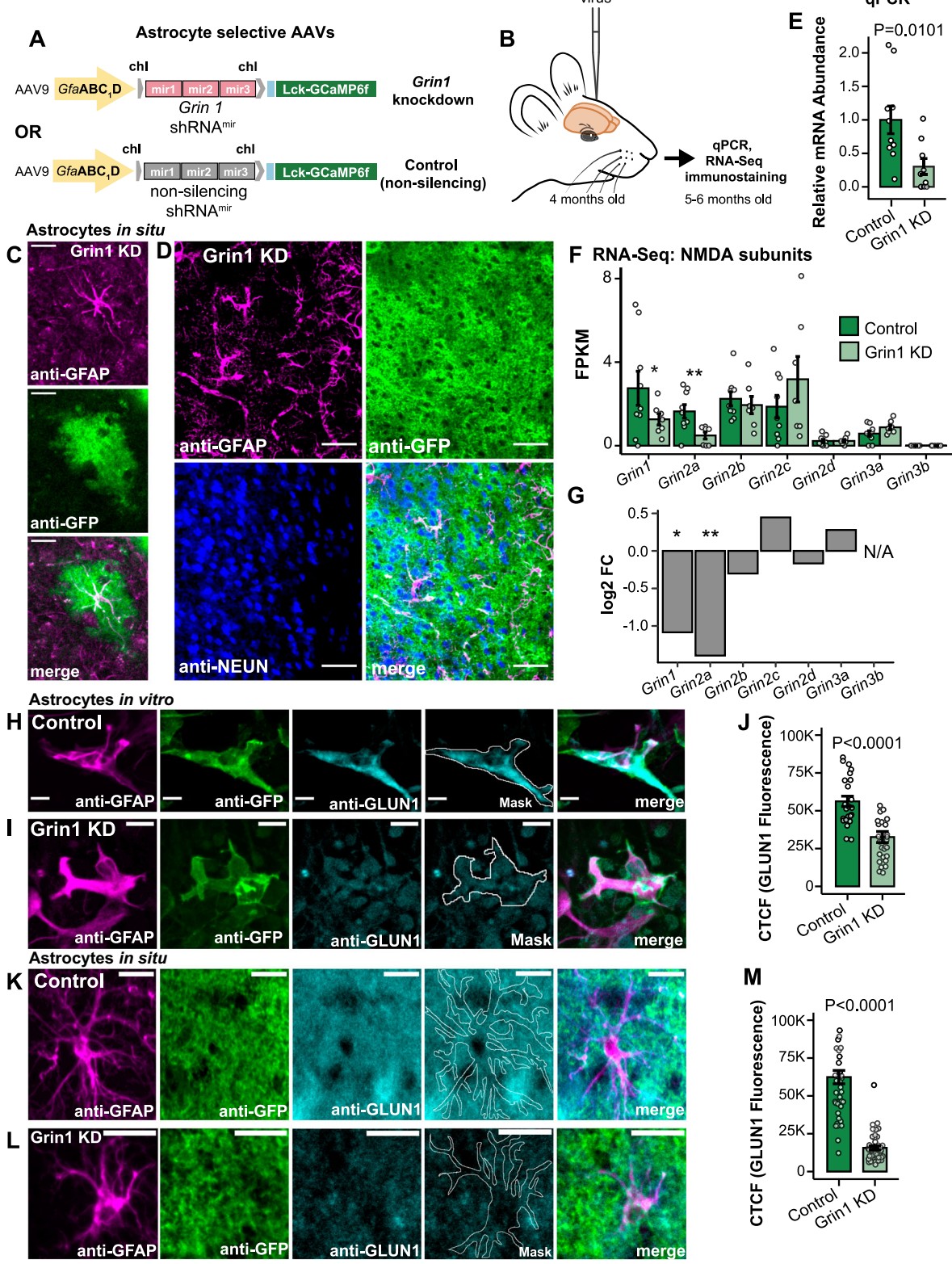

microdomains in control astrocytes, as seen in the whisker stimulation maps in Fig. 3E and Supplementary Movie 1. Control astrocytes had more ROIs during stimulation (9.37 ± 0.72 ROIs/barrel area; Fig. 3G) and higher Ca²⁺ event amplitudes (2.22 ± 0.05 dF/F; Fig. 3H; Supplementary Movie 1). However, in Grin1 KD astrocytes, whisker stimulation failed to increase the number of microdomains (5.08 ± 0.37 ROIs/barrel area; Fig. 3F, G) or Ca²⁺ amplitudes (1.99 ± 0.06 dF/F; Fig. 3H; Supplementary Movie 2). The nature of the sensory stimulus did not

influence this reduction in Grin1 KD astrocyte microdomains, since direct electrical stimulation (400 µA, 4 Hz for 5 s) applied to the whisker pad of anesthetized mice also failed to increase the number of microdomains in Grin1 KD astrocytes (Supplementary Fig. 3A, B). During whisker stimulation, astrocyte microdomains in Grin1 KD astrocytes covered a smaller area, since the fraction of active astrocyte pixels relative to the total astrocyte area was lower (0.07 ± 0.007 control vs. 0.01 ± 0.002 KD; Fig. 3I). Furthermore, the fraction of

**Fig. 1 | Astrocyte-specific knockdown of *Grin1*, the gene for the essential subunit of NMDA receptors. A** Astrocyte-specific AAV9 constructs with three shRNA[mir] targeting *Grin1* for knockdown or three non-silencing shRNA[mir] for control with membrane-tagged Lck-GCaMP6f. **B** AAVs were injected into the whisker barrel cortex and tissue was processed later after viral expression. **C** Lck-GCaMP6f (anti-GFP) from Grin1 KD AAV localized to astrocytes (anti-GFAP) by immunohistochemistry. Scale = 25 µm **D** Astrocytes (anti-GFAP) and not neurons (anti-NeuN) expressed the viral construct. Scale = 50 µm. **E** qPCR for *Grin1* mRNA abundance from isolated astrocytes showed reduced expression in Grin1 KD. Mean ± SEM compared by Mann–Whitney–Wilcoxon (two-sided) test. *n* = 10 control, 9 Grin1 KD mice. **F** NMDA-receptor subunit abundance from RNA-seq in fragments per kilobase per million of mapped reads (FPKM) *P = 0.027, **P = 0.008. **G** Fold change of differential expression of NMDAR subunits from Grin1 KD vs control. *n* = 9 control, 7 Grin1 KD mice *P = 0.027, **P = 0.008. **H, I** Immunocytochemistry of glial cultures transduced with AAV constructs (control and Grin1 KD) and stained for GFAP, GFP (Lck-GCaMP6f), and GLUN1 (the protein of *Grin1*). GFP was used to make a mask for GLUN1 quantification (white outline). Scale bar = 20 µm. **J** The corrected total cell fluorescence (CTCF) for GLUN1 in control and Grin1 KD cultured astrocytes. *n* = 33 control cells from 3 cultures; 30 Grin1 KD cells from 3 cultures. **K, L** Immunohistochemistry of cortical astrocytes transduced with AAV constructs (control and Grin1 KD) and stained for GFAP, GFP (Lck-GCaMP6f), and GLUN1. GFAP was used to make a mask for GLUN1 quantification (white outline). These mice also had *Grin1*[fl/fl] genes and hSYN-Cre AAV virus injection to deplete GLUN1 in Cre-positive neurons. Scale bar = 15 µm. **M** The corrected total cell fluorescence (CTCF) for GLUN1 in control and Grin1 KD astrocytes in tissue depleted of neuronal GLUN1. *n* = 41 control cells from 3 mice; 46 Grin1 KD cells from 3 mice. All bars are mean ± SEM. Statistics for immunostaining were calculated using linear mixed models and Tukey post hoc tests. Source data are provided as a Source Data file.

astrocyte area that was active in more than one stimulation trial was lower in Grin1 KD astrocytes (repeated response score; 0.060 ± 0.005 control vs. 0.036 ± 0.008 KD), suggesting that fewer microdomains were repetitively activated across multiple stimulation trials compared to control astrocytes (Fig. 3E, F, J).

In imaging sessions with SR101, we also extracted $Ca^{2+}$ traces from manually selected somata and end feet along blood vessels. Astrocyte $Ca^{2+}$ microdomains that were identified from our activity-based analysis and did not overlap with the manually selected ROIs were classified as processes. Example traces of $Ca^{2+}$ activity in trials with and without whisker stimulation are plotted in Fig. 3K, L with these different types of astrocyte compartments from FOVs in Fig. 3C, D. When comparing the number of somata, endfeet, or process ROIs that had $Ca^{2+}$ events during stimulation trials, $Ca^{2+}$ signaling was reduced in Grin1 KD astrocyte somata (3.38 ± 0.46 control vs. 1.84 ± 0.18 KD ROIs/barrel area) and process ROIs (31.34 ± 2.73 control vs. 22.42 ± 2.24 KD ROIs/barrel area), but the number of endfeet with $Ca^{2+}$ events was unaffected (6.01 ± 0.80 control vs. 4.19 ± 0.62 KD ROIs/barrel area; Fig. 3M). This suggests that astrocyte NMDA receptors have a greater contribution to $Ca^{2+}$ events in somata and fine processes than endfeet compartments.

## Astrocyte NMDAR knockdown impairs neuronal activity in vivo

To investigate the impact of Grin1 KD on $Ca^{2+}$ events in cortical neurons in vivo, we conducted awake two-photon imaging in the whisker barrel cortex of adult mice injected with a combination of viruses: AAV for neuronal expression of red genetically-encoded calcium indicator, RCaMP1.07 (AAV9-hSYN-RCaMP1.07a) and virus for Grin1 knockdown or control astrocytes (with GFAP-Lck-GCaMP6f; Fig. 4A). We measured RCaMP1.07 $Ca^{2+}$ events as a proxy for neuronal activity in the cortical circuit surrounded by astrocytes with reduced NMDAR expression (Fig. 4B, C). Similar to $Ca^{2+}$ analysis in astrocytes (Fig. 3), neuronal $Ca^{2+}$ ROIs were identified based on their activity (Fig. 4D, E). Active areas primarily localized to somata (identified manually), but also in areas within the "neuropil" that may indicate dendritic activity. Spontaneous neuronal $Ca^{2+}$ events in trials without whisker stimulation occurred more frequently near Grin1 KD astrocytes (for example; Fig. 4E Grin1 KD activity maps vs. Figure 4D control activity maps with no stim; Fig. 4F, G; Supplementary movies 1 & 2). Both the number of active neuronal ROIs per barrel area (8.48 ± 0.8 control vs. 12.42 ± 1.15 KD neuronal ROIs/barrel area) and the $Ca^{2+}$ amplitude (1.23 ± 0.03 control vs. 1.70 ± 0.09 KD vs. dF/F) were higher near Grin1 KD astrocytes compared to controls (Fig. 4H, I), suggesting that the spontaneous neuronal activity was elevated. Upon whisker stimulation, more active neuronal ROIs were detected in control mice (15.76 ± 1.43 ROIs/barrel area, Fig. 4D, J), but the number of active neurons near Grin1 KD astrocytes did not change (10.12 ± 0. 53 ROIs/barrel area; Fig. 4E, G, J), indicating a reduced recruitment of responding neurons during cortical circuit activation. A similar reduced neuronal recruitment was

observed following an electrical whisker pad stimulation (Supplementary Fig. 3C), suggesting that direct electrical stimuli do not overcome neuronal impairments following astrocyte Grin1 KD. Furthermore, these effects were not influenced by sex, as the responses and effect of astrocyte Grin1 KD were the same for neurons and astrocytes from male and female mice (Supplementary Fig. 4). Although there was an overall decrease in the number of responding neuron ROIs near Grin1 KD astrocytes, the mean amplitude of the stimulation-evoked $Ca^{2+}$ events in the remaining responding neurons was larger than controls (1.25 ± 0.02 KD vs. 1.09 ± 0.01 control dF/F; Fig. 4K).

Previous studies have classified neurons based on the amplitude of their $Ca^{2+}$ events into low, mid, and high-responsive cells[39], which reflects their level of recruitment during sensory stimulation. High-responsive neurons have reliable, large amplitude $Ca^{2+}$ events during whisker vibrations[39,40], and this small population of cells "sparsely" encodes sensory stimulation[40-44]. Low and mid-responsive neurons are more weakly recruited during repeated trials of sensory stimulation, but they have a greater capacity for reorganization during changes in sensory input[39]. In order to consider sparse encoding and different neuronal populations, we classified neurons in both Grin1 KD and control mice based on the average neuronal $Ca^{2+}$ amplitude during stimulation, as low-responsive (amplitude < 88th percentile of 1.92 dF/F), mid-responsive (amplitude between 88th percentile of 1.92 dF/F and 98th percentile of 3.47 dF/F) or high responsive (amplitude > 98th percentile of 3.47 dF/F; Fig. 4L). Overall, the calcium amplitudes were the same for each class of neurons near both control and Grin1 KD astrocytes (Fig. 4M), even though the pooled average amplitude for all ROIs was greater near Grin1 KD astrocytes (Fig. 4K). Interestingly, the number of low-responsive neurons was reduced in Grin1 KD mice (14.07 ± 1.17 control vs 9.06 ± 0.49 KD ROIs/barrel area, Fig. 4N), which would account for the decreased number of ROIs per barrel area in the total population (Fig. 4J). Sparse coding of the whisker stimulus was still intact because the number of high-responsive cells was the same between controls and Grin1 KD (4.04 ± 0.46 vs 4.78 ± 0.7 ROIs/barrel area; Fig. 4N), which could explain the elevated average amplitude from the total neuron population in Grin1 KD mice (Fig. 4K).

Neurons in the whisker barrel cortex increase their levels of synchronicity during stimulation[40,45,46]. To estimate synchronization, we calculated the Pearson's correlation coefficient between pairs of neurons in the same FOV in each trial. Improved synchronization occurred during stimulation in neurons near control astrocytes, as the correlation increased during trials with whisker stimulation compared to trials with no stimulus (−0.035 ± 0.007 no stim vs 0.095 ± 0.003 stim; Fig. 5A). For Grin1 KD, the neuronal correlation increased during trials without stimulation and decreased during whisker stimulation (0.152 ± 0.004 no stim vs −0.031 ± 0.005 stim; Fig. 5A). These effects were most pronounced between pairs of high-

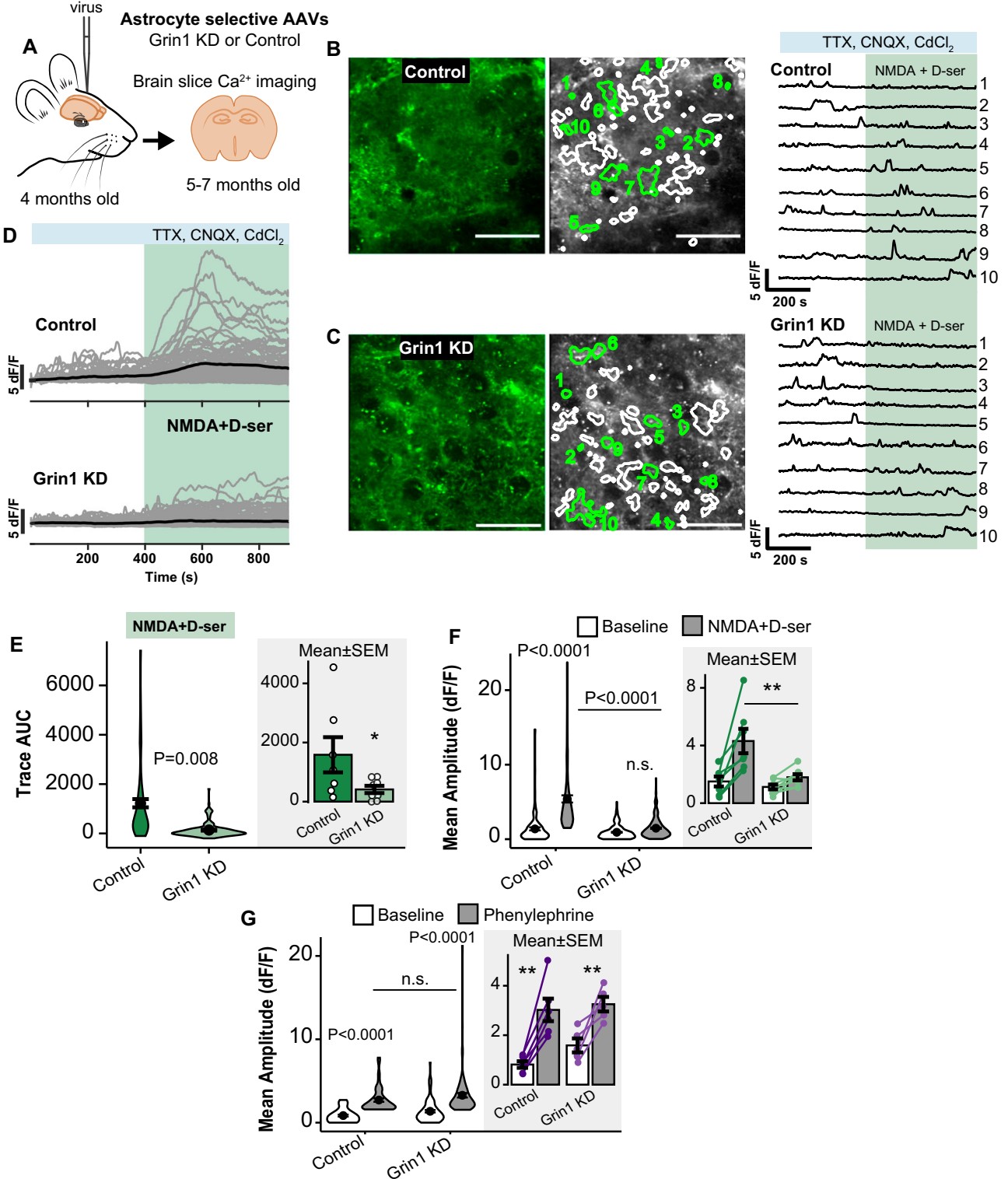

responsive neurons, as their synchronicity increased greatly during whisker stimulation near control astrocytes ($-0.39 \pm 0.003$ no stim vs $0.37 \pm 0.02$ stim; Fig. 5B). However, near Grin1 KD astrocytes, high-responsive neurons had a greater correlation during trials without stimulation ($0.68 \pm 0.003$), reflecting higher spontaneous activity, but the correlation dropped during stimulation ($-0.16 \pm 0.02$; Fig. 5B). This suggests that astrocyte Grin1 KD causes reduced neuronal synchronicity during whisker stimulation that may impact sparse coding and network function.

Neurons adapt their response during prolonged periods of activity, such as long sensory stimulation. We applied an electrical stimulus to the whisker pad (750 μA, 4 Hz) for prolonged periods (30 s) in anesthetized mice transduced with control or Grin1 KD viruses and neuronal RCaMP1.07. Neurons in both control and Grin1 KD mice responded to the stimulus with a sharp $Ca^{2+}$ peak that adapted within a few seconds of the start of stimulation (Fig. 5C). We calculated the slope of this signal decay in $Ca^{2+}$ amplitude and found that control neurons had a more negative decay slope compared to Grin1 KD

**Fig. 2 | A functional reduction of astrocyte Ca²⁺ responses to NMDAR agonists after Grin1 KD. A** Experimental schematic for Grin1 KD or control virus injections and brain slices. **B**, **C** Green fluorescent representative brain slice images of astrocytes transduced with control or Grin1 KD virus (Lck-GCaMP6f; Scale = 50 µm). Regions of interest (ROIs) with active Ca²⁺ events when NMDA (50 µM) + D-serine (10 µM) were applied in the presence of neuronal blockers are indicated as white and green shapes. Example Ca²⁺ traces from 10 ROIs highlighted in green in the middle image are shown. Green shaded area indicates when the NMDA agonists were applied. **D** Individual ROI Ca²⁺ traces (gray) from all brain slices with the mean Ca²⁺ response (black) to agonist application (green) in the presence of neuronal blockers (blue) from control and Grin1 KD slices. **E** Area under the curve (AUC) of Ca²⁺ traces from ROIs during NMDA + D-serine application. Violin plot width shows the frequency of data points in each region and the height shows the distribution of all ROIs. The mean and SEM error bars are also indicated. The bar graph is the average AUC per slice (dots). *$P = 0.046$. **F** Amplitude of Ca²⁺ peaks per ROI during NMDA + D-serine for all ROIs (violin plot) and average per slice (bar graph and dots). **$P = 0.007$. Control: $n = 197$ ROIs from 7 slices from 7 mice, Grin1 KD: $n = 220$ ROIs from 8 slices from 8 mice. **G** Amplitude of Ca²⁺ peaks per ROI during phenylephrine (10 µM) across all ROIs (violin plot) and averaged per slice (dots and bar graph). Control: $n = 47$ ROIs from 6 mice **$P = 0.013$, Grin1 KD: $n = 87$ ROIs from 5 mice **$P = 0.016$. All data are mean ± SEM. Statistics for violin plots were calculated using linear mixed models and Tukey post hoc tests. Stats for bar graphs were calculated using Kruskal–Wallis tests and pairwise Wilcoxon tests (two-sided) with Bonferroni correction. Source data are provided as a Source Data file.

neurons (Fig. 5D), suggesting that they adapted more quickly. This is the first evidence that changes to astrocytic ionotropic signaling can impact and delay neuronal adaptation.

To confirm that the neuronal impairments we observed after Grin1 KD were specific to astrocyte NMDA receptors and not due to a loss of astrocyte calcium signaling in general, we examined neuronal activity in mice lacking the *Itpr2* gene, leading to disrupted IP₃R2 signaling (referred as IP3R2 KO). This mouse strain has fewer astrocyte microdomain Ca²⁺ events due to a reduction in the inositol trisphosphate receptor needed for Ca²⁺ store release from the endoplasmic reticulum[8,10,11]. When considering neuronal RCaMP1.07 activity in our awake paradigm in IP3R2 KO mice, we found minimal alterations in neuronal activity (Supplementary Fig. 5). The level of spontaneous neuron activity was the same between IP3R2 KO and wildtype littermates (Supplementary Fig. 5A). Also, in IP3R2 KO mice, whisker stimulation increased the number of active neurons (with Ca²⁺ activity) compared to trials without stimulation (Supplementary Fig. 5B), unlike Grin1 KD mice where there was no recruitment of responding cells (Fig. 4C). Finally, neuronal synchronization increased during whisker stimulation (larger correlation) in both IP3R2 WT and IP3R2 KO, and in fact, the correlation was slightly greater in IP3R2 KO (Supplementary Fig. 5C), which was different that Grin1 KD (Fig. 5A). This suggests that the neuronal impairments we observed (Figs 4 & 5) were specific to a loss of astrocyte NMDAR signaling.

### Astrocyte NMDAR knockdown reduces fast & delayed-onset microdomains

The majority of astrocyte Ca²⁺ microdomains evoked by sensory stimulation have a delayed signal onset (~5 s after the start of the stimulus[8]) and many of these events are evoked by IP₃R2 signaling and neuromodulators[8]. A population of fast-onset astrocyte Ca²⁺ microdomains that highly correlate with the onset of neuronal activity has also been identified[8,13,31], but the mechanism that underlies these fast signals has not been established[8]. To determine if astrocyte NMDAR activity influences the onset of Ca²⁺ events, we considered the onset latency of neuron and astrocyte Ca²⁺ signals in control and Grin1 KD mice (Fig. 6A). First, the mean onset latency for neurons that responded during whisker stimulation near both control and Grin1 KD astrocytes was similar ($2.53 \pm 0.03$ control vs $2.60 \pm 0.04$ s KD; Fig. 6B). Astrocyte Ca²⁺ microdomains that occurred during whisker stimulation (0–8 s) were categorized as "fast" if their onset latency was less than the median onset latency for the neuronal population and "delayed" if their onset latency was slower than this value[8]. The number of fast microdomains was reduced in Grin1 KD astrocytes compared to controls (Fig. 6C), demonstrating that ionotropic glutamate receptor signaling via astrocytic NMDAR can contribute to fast-onset Ca²⁺ events in astrocytes that potentially rapidly trigger the modulation of nearby synaptic activity. The number of delayed microdomains evoked by stimulation was also lower in Grin1 KD astrocytes (Fig. 6C), suggesting that ionotropic glutamate receptors can also be activated on slower timescales and/ or contribute to metabotropic signaling.

### Astrocyte NMDAR knockdown impairs sensory acuity

Given the impairment of neuronal recruitment, we observed following astrocyte Grin1 KD, we tested the sensory perception of the animal using a whisker touch-based novel texture recognition test[3,47]. In this task, following two days of acclimatization to the testing arena, the whiskers of the mice were trimmed such that only the whiskers corresponding to the virus injection area were long. Mice were habituated to two vertical objects fitted with identical sandpaper texture (e.g., 66 µm particle size) and allowed to explore with their whiskers for 5 min. One object was then replaced with an object covered in a different sandpaper texture (e.g., 98 µm or 267 µm particle size) of the same color, and the time the animal spent exploring the two objects was recorded over 3 min. Mice that can perceive the difference in textures will spend more time with the novel textured object, reflected as a positive discrimination index (DI= (novel texture exploration time – familiar texture exploration time)/ total exploration time). Mice with astrocyte Grin1 KD could not discriminate a small sandpaper grit difference (ΔG = 32 µm; discrimination index near zero) but performed the same as controls when detecting larger grit differences (ΔG = 201 µm; Fig. 7B). This behavior deficit was not due to a memory impairment because Grin1 KD and control mice performed similarly in the novel object recognition test (performed in the same manner as the novel texture recognition), where the objects are more distinct (different colors and shapes) and other cues are used to discriminate and remember objects (Fig. 7A, B).

### ATP rescues neuronal activity & sensory acuity after astrocyte NMDAR knockdown

Since purinergic signaling modulates cortical excitation-inhibition balance[48–52] and ATP is released from astrocytes in vitro after NMDA agonist application[48,53], we hypothesized that astrocyte Grin1 KD could lead to reduced extracellular ATP levels that impair neuronal activity and effect sensory processing. To reverse these effects, we removed the cranial window of Grin1 KD mice and applied ATPγS (50 µM), a less hydrolyzable form of ATP, to the cortical surface for 20 min before awake two-photon imaging or the novel texture recognition test. ATPγS application decreased the number of spontaneously active neurons ($8.487 \pm 0.8$ control vs $12.42 \pm 1.05$ KD vs. $8.486 \pm 0.8$ ATP ROIs/barrel area) and their Ca²⁺ amplitude ($1.23 \pm 0.03$ control vs $1.70 \pm 0.09$ KD vs. $1.28 \pm 0.03$ ATP dF/F) in Grin1 KD to a level similar to neurons in control mice (Fig. 8A, B). ATPγS also increased the number of responding Grin1 KD neurons during whisker stimulation ($8.486 \pm 0.8$ no stim vs $13.45 \pm 1.32$ stim ROIs/barrel area in ATP; Fig. 8C, D) and normalized their amplitude (during stim: $1.09 \pm 0.01$ control vs $1.25 \pm 0.02$ KD vs. $1.15 \pm 0.03$ ATP dF/F; Fig. 8E), rescuing the previous Grin1 KD impairments (Fig. 4). Exogenous ATPγS also recovered the number of low-responsive neurons, while decreasing the number of mid and high-responsive neurons in Grin1 KD compared to controls (Fig. 8F). When considering neuronal synchronization, ATPγS increased the correlation in Grin1 KD mice particularly of those responding to whisker stimulation ($-0.0315 \pm 0.005$ Grin1 KD: vs.

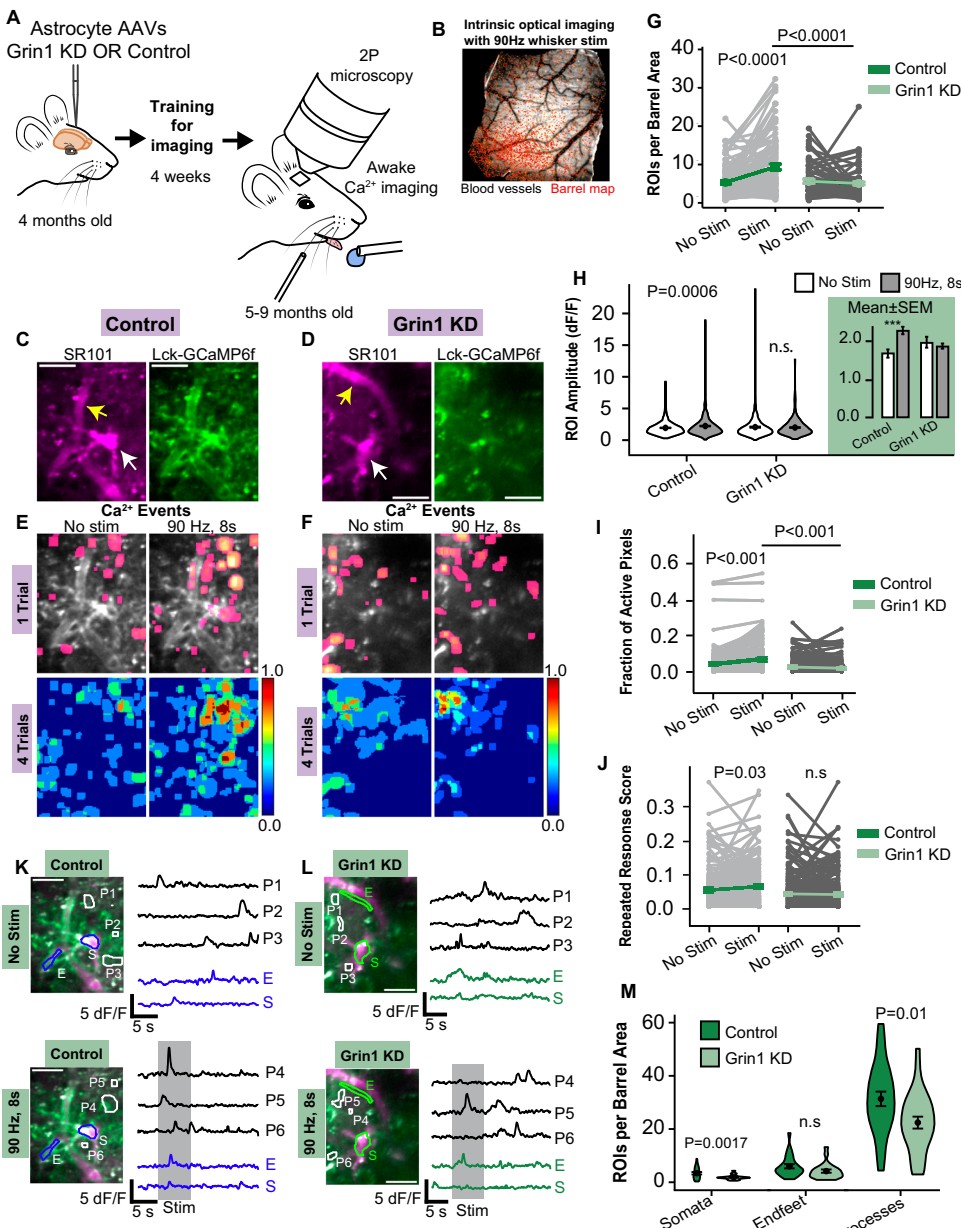

**Fig. 3 | Reduction of stimulation-evoked Ca²⁺ events in Grin1 KD astrocytes in vivo. A** Awake Ca²⁺ imaging of Grin1 KD or control astrocytes after AAV injection. **B** Example whisker barrel map through the cranial window during intrinsic optical imaging. Virus expression was localized within this area. **C, D** Example Lck-GCaMP6f (green) and SR101 (magenta) in L2/3 cortical astrocytes in vivo. Endfeet (yellow arrow) and somata (white arrow) were identified (Scale bar = 25 μm). **E, F** Single trial maps and multiple trial maps (4) of identified active astrocyte Ca²⁺ microdomains from trials with or without whisker stimulation (90 Hz, 8 s). The rainbow scale is the fraction of trials with a response. **G** Number of astrocyte microdomain ROIs with Ca²⁺ events per whisker barrel area was reduced in Grin1 KD during stimulation. Each gray line is a FOV; the mean ± SEM is indicated in color. **H** The mean astrocyte ROI Ca²⁺ amplitude was reduced in Grin1 KD during stimulation. Violin plots show the distribution of all ROIs with the mean and SEM error

bars. The bar graph is the mean ± SEM for all ROIs in the violin plots ***P = 0.006. **I** The fraction of active astrocyte pixels per FOV was reduced during stimulation in Grin1 KD mice. **J** The fraction of astrocyte pixels active in two or more trials (repeated response score) increased with whisker stimulation in control but not Grin1 KD mice. Control: n = 1127 ROIs from 103 FOV in 8 mice, Grin1 KD: n = 448 ROIs from 97 FOV in 11 mice. **K, L** Example traces from no stim and whisker stimulation trials (gray shaded area) for subcellular compartments (S = somata, E = endfeet, and P = processes) with calcium events from cells in (**E**) & (**F**). More regions had peaks during the stimulation time in control astrocytes (Scale bar = 25 μm). **M** The number of astrocyte somata, endfeet, or process ROIs with Ca²⁺ events per whisker barrel area during whisker stimulation. Control: n = 24 FOV from 8 mice and KD: n = 25 FOV from 7 mice. Comparisons made by linear mixed models and Tukey post hoc tests. Source data are provided as a Source Data file.

0.354 ± 0.002 ATP, P = 0.00123; Fig.8G). Lastly, ATPγS restored the sensory acuity of Grin1 KD animals to fine texture differences during the novel texture recognition test (Fig. 8H). As expected, ATPγS application did not improve astrocyte Ca²⁺ signaling or the recruitment of active microdomains during whisker stimulation (Supplementary Fig. 6), suggesting the ATP-related rescue was downstream of astrocyte Ca²⁺ events.

## Discussion

By utilizing an AAV-delivered shRNA^mir knockdown approach to selectively reduce astrocyte NMDARs in the whisker barrel cortex, we show that astrocyte NMDARs contribute to stimulus-evoked microdomain Ca²⁺ events in awake mice. Astrocyte NMDARs are also linked to nearby cortical circuit function, since Grin1 KD reduced and desynchronized responses of neurons to the whisker stimulus and

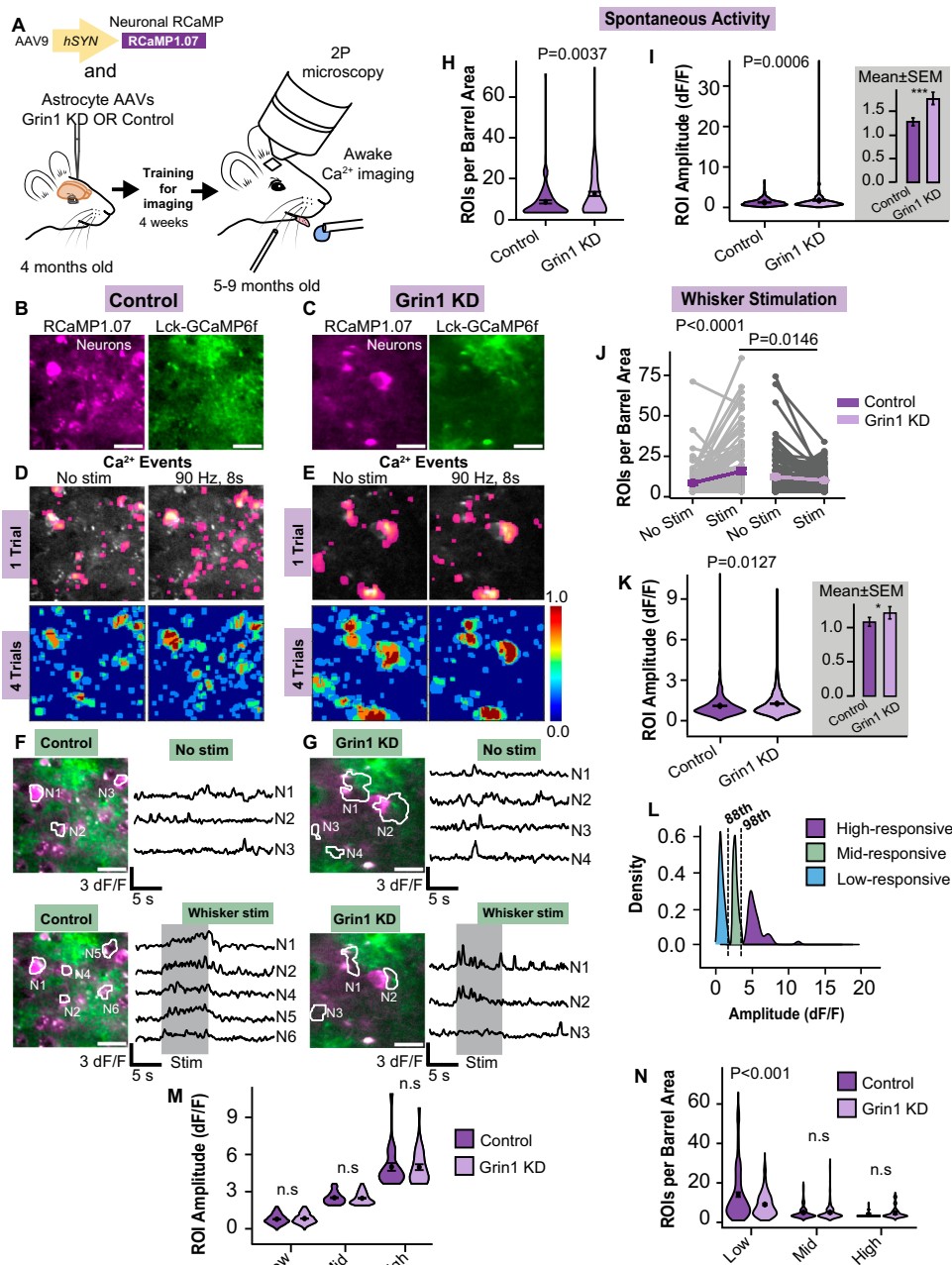

**Fig. 4 | Altered neuronal activity after astrocyte Grin1 KD in vivo. A** Awake neuronal Ca²⁺ imaging near Grin1 KD or control astrocytes. **B**, **C** Example RCaMP1.07 neurons (magenta) and Lck-GCaMP6f astrocytes (green) in L2/3 in vivo (Scale bar = 25 µm). **D**, **E** Single trial and multiple trial Ca²⁺ maps (4) of active neuronal ROIs near Grin1 KD vs. control astrocytes with or without whisker stimulation (90 Hz, 8 s). The color scale is the fraction of trials with a response in that area. **F**, **G** Traces from example neuron ROIs from (**D**) & (**E**) (Scale bar = 25 µm). **H** The number of spontaneous neuronal ROIs per barrel area increased in Grin1 KD mice. **I** ROI Ca²⁺ amplitude in spontaneous neurons increased in Grin1 KD mice. ***P = 0.0006. **J** Whisker stimulation failed to increase the number of active neuron ROIs per barrel area in Grin1 KD. Each gray line is a FOV; the mean ± SEM is indicated in color. **K** ROI Ca²⁺ amplitude was elevated during stimulation in Grin1 KD. Control: n = 1799 ROIs from 122 FOV in 8

mice, Grin1 KD: 1210 ROIs from 144 FOV in 11 mice *P = 0.0127. **L** Classification of neuron types: high-responding, mid-responding, and low-responding neurons based on amplitude percentiles, 88th (1.92 dF/F) and 98th (3.47 dF/F). **M** ROI Ca²⁺ amplitudes for each class of neuron. **N** Number of each neuronal type per barrel area during whisker stimulation. There were fewer low-amplitude neurons in Grin1 KD. High-responsive: n = 35 ROIs from 20 FOV (Control), 37 ROIs from 19 FOV (KD); Mid-responsive: n = 158 ROIs from 58 FOV (Control), 164 ROIs from 69 FOV (KD), Low-responsive: n = 1606 ROIs from 121 FOV (Control), 1009 ROIs from 141 FOV (KD); Mice: Control = 8, Grin1 KD = 11. Violin plots show the distribution of all ROIs or all FOVs with the mean and SEM error bars plotted on top. Bar graphs are the mean ± SEM for all ROIs in the violin plots. Comparisons made by linear mixed models and Tukey post hoc tests. Source data are provided as a Source Data file.

decreased neuronal adaptation. In fact, the effects of astrocyte NMDAR knockdown on the cortical circuit were sufficient to cause a sensory acuity deficit in the animal. These changes in neuronal activity and behavior likely occur due to a loss of purinergic signaling after Grin1 KD, since application of exogenous ATP rescued the Grin1 KD phenotype (Fig. 9). These findings represent a different direction for

astrocyte-neuron interactions in cortical circuits toward ionotropic and purinergic mechanisms.

Previous evidence of the functionality of astrocyte NMDARs has been fraught with controversy because many early studies used primary astrocyte cultures from pups, which may have variable expression levels of NMDAR subunits (for review see[19,22,54]). Also, many other

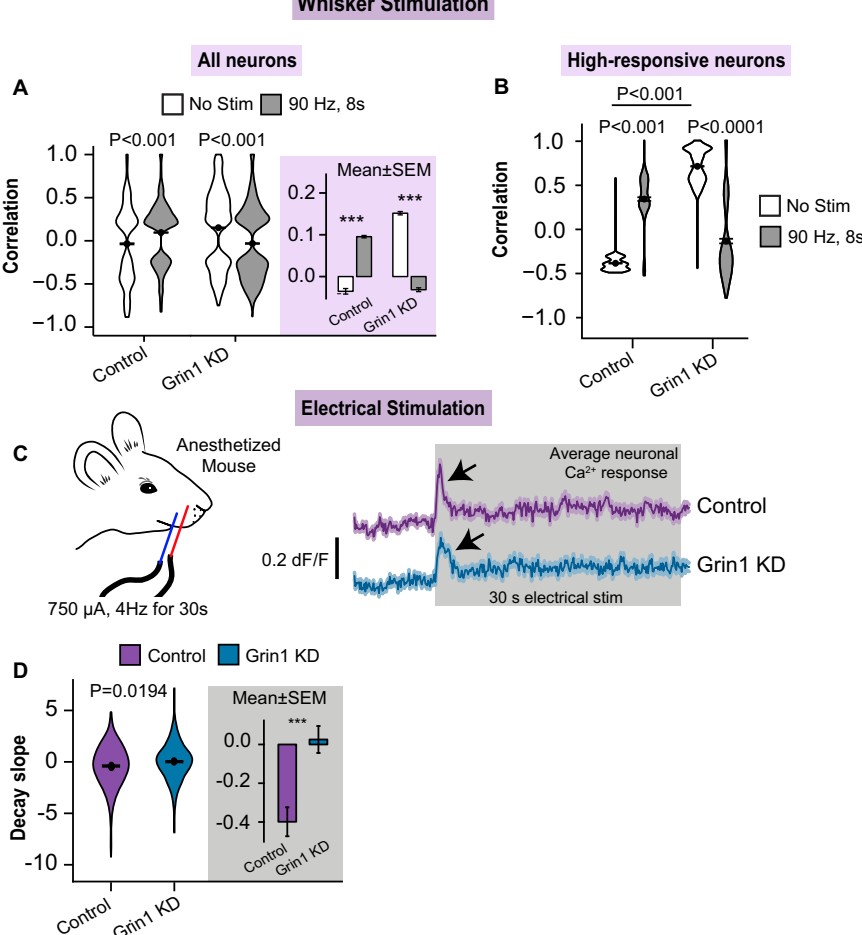

**Fig. 5 | Alterations in neuronal synchronization and adaptation after astrocyte Grin1 KD. A** Neuronal synchronization was determined by Pearson's correlation between pairs of neurons from the same field of view that were spontaneous or stimulation-activated (i.e., Ca²⁺ event during the stimulus period (8 s)) and that had a significant *p*-value during the correlation analysis. Neuronal synchrony increased with stimulation in control but not in Grin1 KD. Purple area is the mean ± SEM for all responding ROI pairs. Control: *n* = 4658 no stim and 16665 stim comparisons from 8 mice; Grin1 KD: *n* = 12825 no stim and 8436 stim comparisons from 12 mice ***P* < 0.001. **B** Neuronal correlation showed that high-responsive neurons were less synchronized in Grin1 KD during whisker stimulation (Control, No Stim: *n* = 1175 comparisons from 15 ROIs, Stim 90 Hz: *n* = 222 comparisons from 31 ROIs, 8 mice). (Grin1 KD, No Stim: *n* = 3266 comparisons from 62 ROIs, Stim 90 Hz: *n* = 367 comparisons from 36 ROIs, 12 mice). **C** Prolonged electrical stimulation of the whisker pad (750 µA, 4 Hz for 30 s) resulted in robust neuronal Ca²⁺ events that quickly adapted (black arrows) for the duration of the stimulus. **D** The slope of the Ca²⁺ event decay was calculated from time 0.5–1 s after the start of the stimulus. The adaptation was slower in Grin1 KD mice. Control: *n* = 714 ROIs from 9 mice; Grin1 KD: *n* = 629 ROIs from 8 mice. Data represented as mean ± SEM ***P* < 0.001. All statistics were calculated using linear mixed model and Tukey post hoc tests. Source data are provided as a Source Data file.

brain cells express NMDARs (neurons, microglia, oligodendrocytes, endothelial cells, etc.[55]), clouding the interpretation of results with bath-applied pharmacology on brain slices or mixed cultures. To target astrocyte NMDARs more precisely, we used a knockdown approach specifically in astrocytes and added a membrane-tagged GCaMP6f to detect Ca²⁺ fluctuations near the plasma membrane where ionotropic receptors are located. We also primarily included adult mice in our study, at an age when astrocyte NMDAR currents are known to be strongest (compared to juveniles[28]). Knockdown of astrocyte NMDARs did not affect the number of spontaneous Ca²⁺ events in astrocytes (Fig. 3), which is similar to results from the hippocampus where NMDAR antagonist, APV, did not block spontaneous Ca²⁺ events in astrocytes of the stratum lucidum[29]. However, we found that astrocyte Grin1 KD prevented the recruitment of Ca²⁺ microdomains during whisker stimulation (Fig. 3). This suggests that NMDARs are not important for spontaneous microdomain activity, but that they are a crucial component of how astrocytes encode sensory information and respond to synaptic glutamate during cortical activity in vivo.

We have previously shown that circuit activation via sensory stimulation can evoke fast-onset Ca²⁺ microdomains in astrocytes on a similar timescale to neurons that are independent of IP₃R2 and Ca²⁺ release from internal stores[8]. After astrocyte Grin1 KD, we found a reduction in the number of fast-onset microdomain Ca²⁺ signals in astrocytes (Fig. 6), confirming that ionotropic receptors can contribute to rapid activity-based dynamics in astrocytes. This also suggests that these receptors have the necessary temporal dynamics to trigger quick feedback mechanisms that alter synaptic activity. It is possible that these fast dynamics occur due to extracellular Ca²⁺ influx[54], and this may occur via several possible mechanisms. First, NMDAR activation could cause direct extracellular Ca²⁺ influx through the receptor channel. Second, the influx of Na⁺ through NMDARs may elevate intracellular Na⁺ inducing reversal of the sodium-calcium exchanger and bringing in extracellular Ca²⁺ in exchange for Na⁺ [16,24,56]. Third, ion influx through NMDARs may depolarize the astrocyte membrane, activating voltage-gated Ca²⁺ channels that permit extracellular Ca²⁺ influx[17]. In addition to Ca²⁺ influx, cortical astrocyte NMDARs may also signal by metabotropic mechanisms, since this has

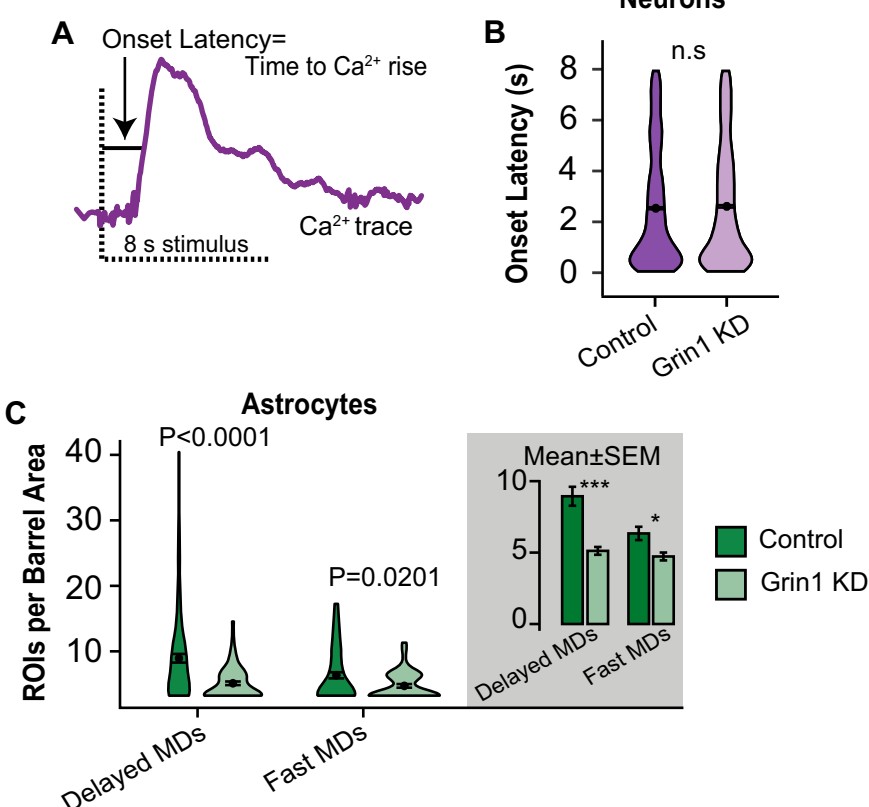

**Fig. 6 | Rapid and delayed-onset astrocyte Ca²⁺ microdomains are reduced in Grin1 KD. A** Onset latency is the earliest time point after the start of whisker stimulation at which the Ca²⁺ signal [dF/F] reached 2.5 SD of the baseline. X-axis = time (s). **B** The onset latency was not different between neurons in control and Grin1 KD neurons. Control: $n = 1799$ ROIs, 8 mice, Grin1 KD: $n = 1210$ ROIs, 11 mice. **C** Astrocyte Ca²⁺ microdomains (MDs) were classified as fast or delayed onset relative to the neuronal onset times. The number of fast-onset and delayed-onset MDs were reduced in Grin1 KD. Gray area is the mean ± SEM for all fields of view (FOV). Control, Delayed MDs: $n = 796$ ROIs from 103 FOV & Fast MDs: $n = 331$ ROIs from 76 FOV, from 8 mice; Grin1 KD, Delayed MDs: $n = 240$ ROIs from 78 FOV & Fast MDs: $n = 193$ ROIs from 56 FOV, from 11 mice. Data are represented as mean ± SEM. All statistics were calculated using linear mixed model and Tukey post hoc tests. *$P < 0.05$, ***$P < 0.001$. Source data are provided as a Source Data file.

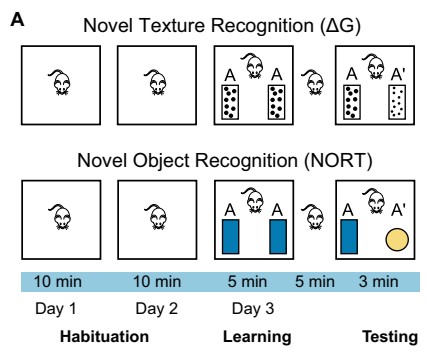

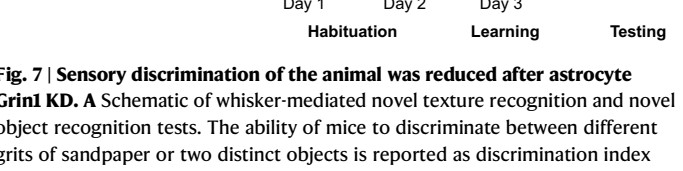

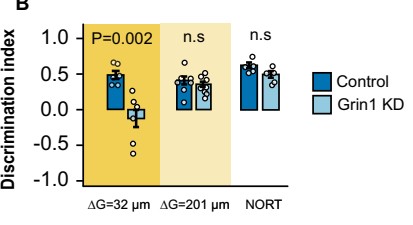

**Fig. 7 | Sensory discrimination of the animal was reduced after astrocyte Grin1 KD. A** Schematic of whisker-mediated novel texture recognition and novel object recognition tests. The ability of mice to discriminate between different grits of sandpaper or two distinct objects is reported as discrimination index (difference in novel – familiar texture exploration time/total exploration time) **B** The discrimination index for small grit difference (ΔG = 32 μm; 150 grit vs. 220 grit; Control mice $n = 6$, Grin1 KD mice $n = 6$), large grit difference (ΔG = 200 μm; 60 grit vs. 220 grit; Control mice $n = 9$, Grin1 KD mice $n = 9$) and the novel object recognition test(Control mice $n = 5$, Grin1 KD mice $n = 5$). Data is mean ± SEM and dots are individual animals. Comparisons were made with the Mann–Whitney–Wilcoxon test (two-sided). Source data are provided as a Source Data file.

previously been described in cultured astrocytes[57,58], and we also observed a reduction in the number of delayed-onset microdomain Ca²⁺ signals that have been shown to involve IP₃R2-mediated metabotropic signaling[8]. Therefore, astrocyte NMDARs are potentially critical for regulating intracellular Ca²⁺ at multiple points from Ca²⁺ influx

to store release, and the precise Ca²⁺ mechanism(s) disrupted in astrocytes during our Grin1 KD approach remain to be determined. It is important to note that Ca²⁺ microdomains were not completely abolished in Grin1 KD astrocytes (Fig. 3), and the amplitude of Ca²⁺ events evoked by phenylephrine application to Grin1 KD slices was similar to

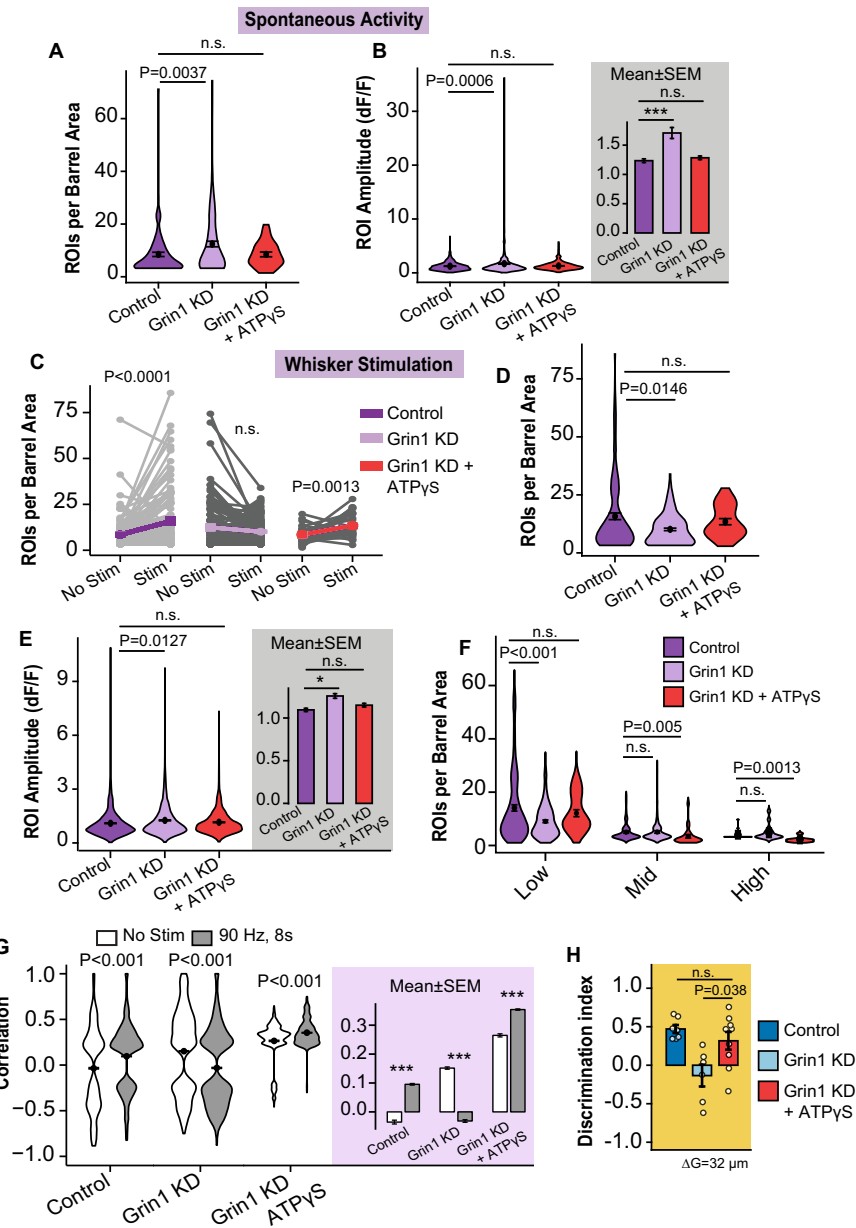

**Fig. 8 | ATPγS application on the cortex rescues the neuronal and behavioral effects of astrocyte Grin1 KD in vivo. A** ATPγS increased the number of spontaneous neuronal ROIs per barrel area in Grin1 KD mice. **B** ATPγS reduced the Ca²⁺ amplitude in spontaneous neurons in Grin1 KD mice. Gray area is the mean ± SEM for all ROIs ***P = 0.0006. **C** ATPγS increased the number of neuronal ROIs per barrel area responding to whisker stimulation compared to trials without stimulation in Grin1 KD. Each gray line is a FOV; the mean ± SEM is indicated in color. **D** The number of responding neurons during whisker stimulation in the ATPγS treated group was not different than controls. **E** ATPγS reduced the Ca²⁺ amplitude of Grin1 KD neurons during stimulation. Gray area is the mean ± SEM for all ROIs *P = 0.0127. **F** ATPγS treatment in Grin1 KD mice caused a redistribution of the neuronal population responding to whisker stimulation: The number of low-

responsive neurons increased during whisker stimulation to a number similar to controls, while the number of mid-responsive and high-responsive neurons decreased. For Fig. 8 A–F, N = 54 FOVs from 10 mice in the ATPγS group; other groups from Fig. 4. **G** The correlation of Grin1 KD neurons improved with ATPγS, where the correlation between neurons in the same field of view was elevated, particularly during whisker stimulation. Grin1 KD + ATPγS n = 1555 no stim and 3661 stim comparisons from 10 mice. **H** The texture discrimination and sensory acuity of Grin1 KD mice during the novel texture recognition test was improved by ATPγS. Control mice n = 6, Grin1 KD mice n = 6, Grin1 KD + ATPγS mice = 9. Violin plots show the distribution of all ROIs or all FOVs with the mean and SEM error bars plotted on top. All statistics were calculated using linear mixed model and Tukey post hoc tests. ***P < 0.001. Source data are provided as a Source Data file.

controls (Fig. 2G). This suggests that astrocyte Ca²⁺ and GPCR signaling were still intact after Grin1 KD.

NMDAR have been linked to sensory information processing[59,60], including Bayesian inference[61], but we provide the first evidence that NMDARs on astrocytes contribute to this type of behavioral processing. We observed reduced sensory acuity in our novel texture recognition task (Fig. 7), which can be attributed to several aspects of

the changes in neuronal activity we described following astrocyte Grin1 KD (Figs 4, 5). First, the balance of cortical excitation-inhibition is essential for the efficient processing of sensory information. In quiet, awake animals, there is low pyramidal neuron and high interneuron activity, which generates a blanket of inhibition that decreases the "noise"[42,62]. Whisker deflection evokes high contrast activity compared to baseline firing during quiet wakefulness, and opens up local

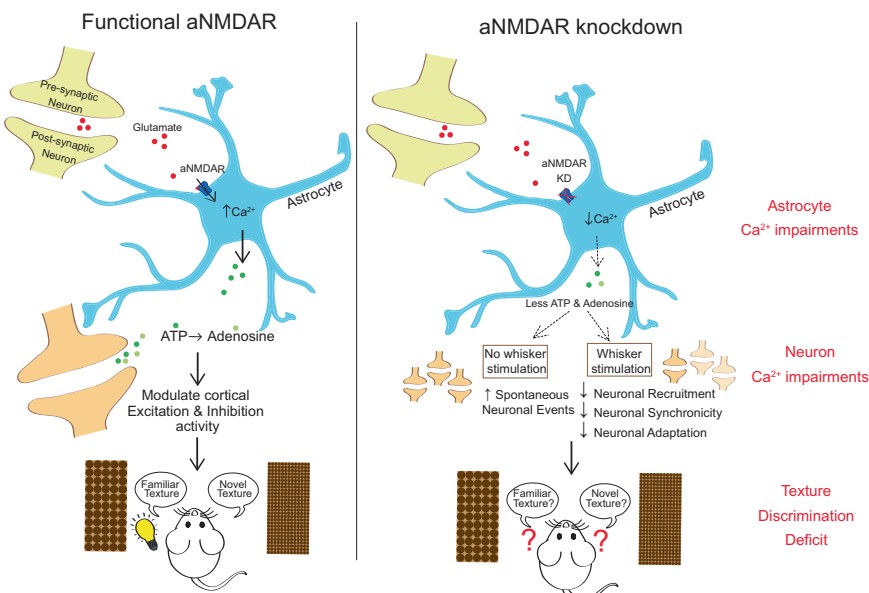

**Fig. 9 | Summary diagram of cortical alterations after astrocyte NMDA-receptor knockdown.** Under physiological conditions, astrocyte NMDA receptors (aNMDAR) may trigger purinergic release necessary for the balance of cortical excitation and inhibition that optimizes the acuity of sensory discrimination (as seen in a whisker-mediated discrimination task). Upon knockdown of *Grin1*, encoding NMDA-receptor subunit GLUN1, astrocyte Ca²⁺ signaling is reduced and may disrupt purinergic release. This elevates neuronal excitability at rest, while decreasing neuronal recruitment, synchronization, and adaptation during sensory stimulation. This loss of purinergic signaling also impairs the sensory perception of the animal. Supplying exogenous ATP is enough to rescue these knockdown-induced impairments (Fig. 8).

transient "signals" (i.e., holes) in the blanket of inhibition[62]. We found that spontaneous neuronal activity during quiet wakefulness was higher near Grin1 KD astrocytes and that fewer neurons responded during whisker stimulation (Fig. 4) with less synchronization (Fig. 5). Thus, astrocyte Grin1 KD disrupts the blanket of inhibition during quiet wakefulness increasing the "noise" at baseline, which suppresses the contrast or signal-to-noise ratio of whisker deflection-evoked responses needed for reliable, efficient stimulus representation and optimal sensory perception[62].

Second, high-responsive neurons are the primary cells that encode sensory information[40–44], and create holes in the blanket of inhibition. We found that the number of high-responsive cells in Grin1 KD mice was the same as control (Fig. 4L); however, these cells were less synchronized during whisker stimulation (Fig. 5B). This indicates network dysfunction and impaired encoding of the sensory stimulus, which could occur as a result of the elevated baseline "noise" or due to direct effects of astrocyte NMDAR KD on these high-responsive cells. Such desynchronization could also mediate reduced sensory processing and acuity[63].

Third, low-responsive neurons are recruited during plasticity[39,41,64] and play a role in encoding stimuli after a change in sensory experience[39]. We found a reduction in the recruitment of low-responsive neurons after astrocyte Grin1 KD (Fig. 4L). Although the contribution of low-responsive cells to network activity during sparse firing has been suggested to be marginal[41], our results indicate that they could be important for normal sensory processing, perhaps through plasticity mechanisms[39]. Hippocampal astrocyte NMDARs mediate heterosynaptic plasticity[17,27], and similar mechanisms may occur in cortical circuits. However, further experiments are required to determine if a reduction in cortical astrocyte NMDARs following our Grin1 KD approach alters cortical plasticity.

Lastly, cortical neurons strongly adapt to prolonged trains of stimuli due to short-term depression at thalamocortical synapses[65,66], and adaptation is beneficial for sensory processing because the coding efficiency is enhanced permitting better discrimination of similar stimuli[67,68]. When we applied prolonged electrical stimuli to the whisker pad of anesthetized mice, we found that neurons near Grin1 KD astrocytes adapted slower, suggesting their adaptation response was impaired. Thus, it is perhaps unsurprising that Grin1 KD mice had difficulty discriminating similar textures (Fig. 7). At this stage because our Grin1 KD virus is expressed across multiple cortical layers, it is not possible to determine if impairments in adaptation occur due to disrupted depression at thalamocortical synapses in deeper cortical layers or local changes in the L2/3 microcircuit. Future approaches will help to disentangle these changes and better link deficits in Grin1 KD adaptation to altered perception of similar stimuli[67,68].

Exogenous ATPγS rescued the neuronal and sensory acuity impairments after astrocyte Grin1 KD (Fig. 8), demonstrating that the impacts of a loss of astrocyte NMDARs is reversible (i.e., Grin1 KD does not cause permanent circuit damage). ATP modulates both excitatory and inhibitory neuron activity[69]. In our experiments, increasing the levels of ATP in Grin1 KD mice reduced spontaneous neuronal activity and baseline "noise". ATP also improved neuronal recruitment during whisker stimulation (especially low-responding neurons) and neuronal synchronization, which likely enhanced the "signal" for more efficient sensory encoding[63]. As discussed above, these properties optimize sensory perception[62], which could explain why ATP improved sensory acuity in our whisker discrimination task. Interestingly, we found that ATPγS treatment caused a loss of mid/high-responding neurons in Grin1 KD (Fig. 8F), which raises the possibility that there is a network redistribution that converts high/mid-responding neurons to low-responsive cells.

Astrocytes release ATP[48–50], including after NMDAR agonist application to slices or cultured cells[48,53]. Since our results suggest that astrocyte Grin1 KD causes a loss of extracellular purines, it is tempting to speculate that ATP gliotransmission is disrupted (Fig. 9). In the cortex, a loss of astrocyte-derived ATP has been linked to autism and depressive-like behaviors in mice via reduced GABAergic transmission[49,50]. Cortical astrocyte ATP can facilitate enhancement of pre-synaptic GABA release[48]. Furthermore, ATP is metabolically converted into adenosine by ectonucleotidases, and cortical adenosine

can inhibit excitatory pyramidal neuron activity[52]. Given that ATPγS was applied for 20 min before the start of our experiments, there could have been some slow degradation to adenosine during our recordings. Thus, ATP and adenosine may both have contributed to the rescue of Grin1 KD by shifting the cortical balance toward inhibition for the enhancement of sensory encoding. Future directions may directly link cortical astrocyte NMDAR activity with ATP release in vivo and disentangle the contribution of ATP vs adenosine at cortical synapses to better understand how astrocyte-neuron interactions via astrocyte NMDARs regulate neural network gain control for optimal synchrony and information processing[7].

In conclusion, these data (1) enhance our understanding of astrocyte NMDARs in cortical circuits, a topic that has not yet been explored clearly in vivo and (2) provide important insights into how astrocytes influence mouse whisker barrel microcircuits needed for sensory acuity. We propose that astrocytes respond to glutamatergic transmission with Ca$^{2+}$ microdomains evoked by NMDAR activity. This Ca$^{2+}$ response triggers astrocyte-neuron feedback via purinergic signaling that regulates neuronal activity needed for accurate sensory processing (Fig. 9).

## Methods
### Mouse models
All experimental procedures outlined below were approved by the Animal Care Committee of the University of Manitoba in accordance with the Canadian Council on Animal Care (protocols 18-025 & 22-024). Mouse strains, sexes, and ages for each experiment are listed in Table 1. C57BL/6NCrl, IP3R2 WT, and IP3R2 KO (Itpr2tm1Chen;[70]) were housed under a standard 12-h light/dark cycle with *ad libitum* access to food and water (unless water was removed overnight for awake two-photon imaging and training). The average temperature and humidity of the mice housing room was 20.4 °C and 40%, respectively. For neuronal GLUN1 depletion, mice with *Grin1*$^{fl/fl}$ (B6.129 Grin1$^{tm2Stl/tm2Stl}$; Jax #005246[34]) and EYFP$^{fl/fl}$ (Gt(ROSA)26Sor$^{tm1(EYFP)}$; Jax #006148[35]) were used for virus injections. These animals also had an inducible Cre recombinase in microglia (Cx3cr1$^{tm2.1(cre/ERT2)Jung;}$ JAX #020940), but tamoxifen was not given to the animals. CD1 mouse pups were used for glial cultures. At the end of experiments, mice were either (a) administered a lethal dose of pentobarbital (>150 mg/kg) for transcardial perfusion and tissue collection or (b) anesthetized with isoflurane in a bell jar and decapitated for brain slice imaging experiments.

### Strategy for *Grin1* knockdown and virus production
The AAV9-hSYN-RCaMP1.07 was packaged by the Vector Core at the University of North Carolina at Chapel Hill and has been used previously in vivo[8]. The shRNA$^{mir}$ silencing and non-silencing constructs were custom-designed. Multiple online siRNA prediction sites were used to search for optimized targeting sequences within the mouse *Grin1* coding sequence to identify target regions that were consistent between different algorithms. In the end, 3 target regions, 5′-GTTGAGCTG-TATCTTCCAAGAG-3′, 5′-CTATAGTTGGCAAACTTCCGGT-3′, 5′-CTTG ATGAGCAGGTCAACGCAG-3′, were chosen that include the 5′ UTR, the ORF and the 3′ UTR. The 19nt targets were extended 5′ and 3′ to a final length of 22 nucleotides and modified to a sense and antisense miR-30-based shRNA sequence. A non-silencing hairpin derived from the pGIPZ library was used as our control. All sequences were cloned in silico into the optimized miR-E backbone[71], and subsequently linked together to form a chain of 3 hairpins, separated by a spacer sequence. This shRNAmir-E multimer was encoded within a chimeric intron (chI) sequence between the splice donor and splice acceptor branch point sites. This entire chI-[3X(shRNAmir-E)] cassette, along with cloning sites, was synthesized using the ThermoFisher GeneArt Gene Synthesis service. The shRNAmir-E cassettes were digested out of the GeneArt constructs and cloned into a pssAAV-2-sGFAP-Lck-GCaMP6f backbone, generating the pssAAV-2-sGFAP-chI-[3X(shRNAmir-E)]-Lck-GCaMP6f. Ligated constructs were transformed into MDS42 *E. coli* cells by heat shock and grown in Terrific Broth (TB) + carbenicillin with vigorous shaking at 37 °C for 16 h. Plasmids were isolated using the PureLink HiPure Plasmid Maxiprep kit (ThermoFisher Scientific), eluted in sterile water, and sequence confirmed (3′ end of sGFAP promoter, entire hairpin cassette, and Lck-GCaMP6f). These silencing and non-silencing constructs were packaged in AAV9; hGFAP-chI[3x(shmGrin1)]-Lck-GCaMP6f (v987-9) and hGFAP-chI[3x(shm/rNS)]-Lck-GCaMP6f (v372-9), by Viral Vector Facility, University of Zurich, and are available for order in their repository.

## Table 1 | Mouse sex and age used in the study

| Experiment | Number & sex | Age at the time of surgery | Age at the time of experiment |
|---|---|---|---|
| FACS, qPCR, RNA-seq | 10 Control, 9 Grin1 KD All Female | 4–5 months | 5–6 months |
| Immunohistochemistry | Virus quantification. (Fig. 1C, D and Supplementary Fig. 1) 6 Control/ 6 Grin1 KD 3 Males/ 3 Females | 4 months | 5–6 months |
| | GluN1 Depletion. (Fig. 1K–M, Supplementary Fig. 2) *Grin1*$^{fl/fl}$ x EYFP$^{fl/fl}$ 3 Control/ 3 Grin1 KD All Female | 3 months | 4 months |
| Brain slice Ca$^{2+}$ imaging | 7 Control; 3 Males/ 4 Females 8 Grin1 KD; 3 Males/ 5 Females | 4–5 months | 5–8 months |
| In vivo awake Ca$^{2+}$ imaging | 8 Control; 4 Males/ 4 Females 11 Grin1 KD; 5 Males/ 6 Females | 3–5 months | 4–6 months |
| Novel texture recognition task | 9 Control; 4 Males/ 5 Females 9 Grin1 KD; 5 Males/ 4 Females | 3–5 months | 7 months |
| Novel objective recognition task | 5 Control; 2 Males/ 3 Females 5 Grin1 KD; 3 Males/ 2 Females | 4 months | 6 months |
| ATPγS experiments: Ca$^{2+}$ imaging and novel texture recognition test | 10 Grin1 KD; 5 Males/ 5 Females | 4 months | 5–6 months |
| IP3R2 knockout mice | IP3R2 WT; 3 Females IP3R2 KO; 4 Females | 4 months | 5–6 months |

## Virus injection surgery

For some experiments, such as brain slice Ca²⁺ imaging or neuronal GLUN1 depletion, mice received an intracortical AAV virus injection at least 1 month before experiments. Briefly, animals were anesthetized with isoflurane (4% induction, 1.5–2% maintenance) and fixed in a stereotaxic frame (RWD; Model 68507). An incision was made over the left hemisphere to expose the skull. A small hole was made over the whisker barrel cortex with a dental drill (approximately 3.5 mm lateral, 1.5 posterior to bregma[72]). A glass micropipette (pulled with Sutter Instruments; P97) was inserted through the hole into the cortex by a custom-made hydraulic pump at a rate of 100 µm/min up to 400 µm deep. AAV virus AAV9-sGFAP-Grin1-shRNA$^{mir}$-Lck-GCaMP6f ($1 \times 10^{12}$ particles/ml) or AAV9-sGFAP-NS-shRNA$^{mir}$-Lck-GCaMP6f ($1 \times 10^{12}$ particles/ml) was injected at 50 nL/min until 400 nL was injected. For depletion experiments, AAV9-hSYN-mCherry-Cre (from University of Zurich Viral Vector facility; v230-9; $6.1 \times 10^{11}$ particles/ml) was mixed with the astrocyte viruses before injection. The hole was sealed with dental acrylic (Ivoclar; Tectric EVOFlow) polymerized with blue light and the skin was sutured. Mice received meloxicam (2 mg/kg s.c.) every 24 h for several days after the surgery until recovery, and buprenorphine slow release (0.5 mg/kg s.c.) every 72 h over 3 days.

## Headpost-implantation surgery

The cranial window surgery was conducted as previously described[8], in two separate surgeries 48–72 h apart. First, animals were anesthetized with isoflurane (4% induction, 1.5–2% maintenance) and were fixed in a stereotaxic frame (RWD; Model 68507). A headcap was fitted on the skull using one layer of bonding agent (Bisco Dental; All Bond Universal Adhesive) and a few layers of dental cement (Ivoclar; Tectric EVOFlow) polymerized with blue light. A custom-made aluminum head post was attached to the headcap at the back of the head. The headcap covered all areas of the skull except for the left somatosensory cortex which was later used for craniotomy and virus injection. Animals were given 5% glucose (0.3 ml s.c.) to aid recovery. Animals also received meloxicam (2 mg/kg s.c.) every 24 h for several days after the surgery until recovery, and buprenorphine slow release (0.5 mg/kg s.c.) every 72 h over 6 days.

## Intrinsic optical imaging

Two days after headpost-implantation surgery, the somatosensory cortex was mapped using intrinsic optical imaging (IOI) through the skull to localize specific whisker areas and identify proper regions for virus injection. Animals were anesthetized with isoflurane (4% induction and 0.5–1% maintenance) and head-fixed using the implanted headpost. The skull was washed with sterilized cortex buffer (NaCl, 125 mM; KCl, 5 mM; glucose, 10 mM; HEPES, 10 mM; CaCl₂, 2 mM; MgSO4, 2 mM; pH ~7.4) to visualize the vasculature. The solution was then replaced with ultrasound gel (HealthCare Plus) and a small cover slip (5 mm diameter; Fisher Scientific) was placed on top. Images were acquired using a 12-bit CMOS camera (Basler Ace acA2040-55 µm) focused 400 µm below the cortical surface, under 630 nm red light illumination. Whisker stimulation (90 Hz, 10 s), by lateral deflection of a single whisker threaded into a glass capillary affixed to a piezo actuator (PiezoDrive), increased the blood flow to the corresponding area of the somatosensory cortex which was identified by increased light absorption.

## Cranial window surgery

On the second step of the surgery, following IOI, animals were anesthetized with triple-anesthesia including fentanyl (0.05 mg/kg), medetomidine (0.5 mg/kg), and midazolam (5 mg/kg) injected subcutaneously. A craniotomy was cut over the barrel cortex. According to the IOI map, using a glass micropipette (Sutter Instruments; P97), and a custom-made hydraulic pump, 300 nL of virus mix was injected at 50 nL/min rate at a depth of 400 µm into the responding whisker areas. A mixture of AAV9-sGFAP-shRNA-Lck-GCaMP6f ($1 \times 10^{12}$ particles/ml) and AAV9-hSYN-RCaMP1.07 ($2.4 \times 10^{12}$ particles/ml) for Grin1 KD or AAV9-sGFAP-NS-shRNA-Lck-GCaMP6f ($1 \times 10^{12}$ particles/ml) and AAV9-hSYN-RCaMP1.07 ($2.4 \times 10^{12}$ particles/ml) for control animals. A square sapphire glass ($3 \times 3$ mm) was lightly pressed on the exposed brain using a stereotaxic arm and fixed with dental cement to the head cap. Animals were given 5% glucose (0.3 ml s.c.) to help recovery, and anesthesia antagonist (flumazenil and atipamezole, 0.5 mg/kg and 2.5 mg/kg, s.c.). Animals received meloxicam (2 mg/kg s.c.) every 24 h for several days after the surgery until complete recovery. Mapping of the barrel cortex (IOI) was repeated through the cranial window two weeks after surgery and before two-photon imaging commenced.

## Behavior training for awake imaging

One week after surgery, training started for awake two-photon imaging. Animals were handled twice a day for 3–5 days until they were comfortable being handled by the experimenter. Then, they were introduced to the head fixation multiple times a day for 3–4 days by restraint inside a custom apparatus tube with a holder for the implanted head post. Restraint started from several seconds and increased to several minutes as the animal became accustomed to the setup. Finally, when animals were acclimatized to the head fixation tube, they were water-deprived overnight, and they were presented with water from a lick spout while restrained and their whiskers were periodically stimulated. Starting with short trials (12.5 s), a water drop was presented simultaneously with an auditory cue at the end of each trial (10th second). The aim was to train the animal to sit still, accept whisker stimulus during the trial, and receive water as a reward. The whisker stimulus was achieved by threading a single whisker into a glass capillary affixed to a piezo element (PiezoDrive) vibrated at 90 Hz. The length and the number of trials were increased to 25 s and up to 50 trials per session, as the animal showed signs of being accustomed to the setup. Animals were trained in two sessions a day, 3–5 days a week, for a total time of 2–4 weeks depending on their performance.

## In vivo two-photon Ca²⁺ imaging

Awake, water-deprived animals were imaged while head-restrained in the water-reward task setup under a two-photon laser-scanning microscope (Ultima In Vivo, Bruker Fluorescence Microscopy) with a 20× water immersion objective (N20X-PFH 20X Olympus XLUMPLFLN Objective, 1.00 NA, 2.0 mm WD). RCaMP1.07 and Lck-GCaMP6f were excited at 990 nm with a Ti:sapphire laser (Coherent, Ultra II) and emission light was split with a 565 LP dichroic to GaAsP photomultiplier tubes with 595/50 nm band pass filter (red) or a 525/70 band pass filter (green). At the start of each imaging session, fields of view in cortical layer 2/3 (depth 110–280 µm) were identified based on fluorescence expression and the corresponding IOI barrel map for each whisker. Short trials (12.5 s) with 3 s of whisker stimulation (90 Hz) were used to confirm that neurons responded to whisker stimulation and that the correct whisker was stimulated. Once the cells were selected, a high resolution ($512 \times 512$ pixels, 1.17 fps) image was collected of the field of view as a reference (to avoid imaging same fields of view across multiple sessions). Then, data was acquired at $128 \times 128$ pixels (13.84 fps) for 25-s trials including 5 s of baseline, 5 s of whisker stimulation (90 Hz) followed by 12 s without stimulation. For each field of view, a total of 10 trials were performed including 5 trials without stimulation alternating with 5 stimulation trials. The whisker stimulus was presented by threading whiskers into a glass capillary affixed to a piezo element vibrated at 90 Hz. Animals were imaged 3–5 days a week for up to 2 months.

For experiments with electrical whisker pad stimulation, animals were anesthetized with isoflurane (1.5%) and two electrodes were implanted superficially in the skin within the whisker area. Mild electric

stimuli (4 Hz, 1 ms pulse every 249 ms) were applied at 400 or 750 μA from 5 or 30 s. Images on the two-photon microscope were acquired in the same manner as awake imaging above.

For experiments considering astrocyte compartments (somata, endfeet, etc.; Fig. 3), animals were briefly anesthetized with isoflurane and SR101 was injected into the tail vein (20 mg/kg)[38]. Ninety minutes later after SR101 had crossed the blood-brain barrier and entered astrocytes, awake two-photon imaging was conducted.

For experiments with ATPγS, mice were anesthetized with isoflurane and the cranial window was removed by drilling away the surrounding dental cement. The dura was disrupted in one corner of the exposed window to facilitate drug penetration. ATPγS (50 μM) was applied to the exposed cortical surface in aCSF. After a 20 min incubation, agarose (1%) was applied to the surface of the brain and a temporary glass coverslip was cemented in place on top. Animals were allowed to recover from anesthesia and then they were placed under the two-photon microscope or in the arena for the novel texture recognition test.

### Brain slice two-photon Ca²⁺ imaging

Animals were anesthetized with isoflurane, cervically dislocated, and decapitated. Brain was removed and rapidly placed in ice-cold carbogen-saturated (95% oxygen; 5% carbon dioxide) slicing buffer (N-methyl-D-glucamine, 93 mM; KCl, 3 mM; $MgCl_2$ * $6H_2O$, 5 mM; $CaCl_2$ * $2H_2O$, 0.5 mM; $NaH_2PO_4$, 1.25 mM; $NaHCO_3$, 30 mM; HEPES, 20 mM; glucose, 25 mM; sodium ascorbate, 5 mM; sodium pyruvate, 3 mM). The hemisphere injected with virus was cut and mounted on a vibratome with slicing chamber containing ice-cold oxygenated slicing buffer. Sagittal slices of 300 μm thickness were cut and incubated in oxygenated 32 °C recovery solution (NaCl, 95 mM; KCl, 3 mM; $MgCl_2$ * $6H_2O$, 1.3 mM; $CaCl_2$ * $2H_2O$, 2.6 mM; $NaH_2PO_4$, 1.25 mM; $NaHCO_3$, 30 mM; HEPES, 20 mM; glucose, 25 mM; sodium ascorbate, 5 mM; sodium pyruvate, 3 mM) for 1 h.

Astrocyte Ca²⁺ events were recorded in oxygenated aCSF (NaCl, 125 mM; KCL, 2.5 mM; $NaH_2PO_4$, 1.25 mM; $MgCl_2$, 1 mM; $CaCl_2$, 2 mM; $NaHCO_3$, 25 mM; glucose, 25 mM) at 35 °C using Ultima In Vitro Multiphoton Microscope (Bruker Fluorescence Microscopy) with a 40× water immersion objective (Olympus). Lck-GCaMP6f was excited at 930 nm and a high resolution (512 × 512 pixels) scanning of the slice was done to identify Lck-GCaMP6f-expressing astrocytes. Once the recording area was determined, a cocktail of neuronal activity blockers including TTX (1 μM), CNQX (10 μM), and $CdCl_2$ (100 μM) was bath-applied to the slice for 20 min. Then, images (256 × 256 pixels, 2.58 frames per second (fps)) were recorded for 15 min while NMDA (50 μM) and D-serine (10 μM) were applied for the first 5 min and then washed out with aCSF for the next 10 min. After washing for 20 min, another 15-min image with the same speed and resolution was recorded while phenylephrine (PE) (10 μM) was applied for the first 5 min and washed out using aCSF for the next 10 min.

### Two-photon image analysis

Image analysis was done as previously described[8], using ImageJ and MATLAB R2020b (MathWork). For in vivo data analysis, neuronal somata regions of interest (ROIs) were hand-selected using ImageJ based on the RCaMP1.07 fluorescence. ROIs within astrocytes processes (in vivo and in vitro image analysis) or neuronal dendrites (in vivo image analysis) were automatically identified using an activity-based algorithm from a custom-designed image processing toolbox for MATLAB (Cellular and Hemodynamic Image Processing Suite (CHIPS)[36]. Active pixels were defined based on two criteria relative to a sliding temporal boxcar of 5 s across the movie: (1) amplitude changes-active pixels exceeded 7 times the standard deviation (in vivo images, Lck-GCaMP6f and RCaMP1.07) or 5 times the standard deviation (in vitro images, Lck-GCaMP6f) of the mean pixel intensity in this temporal boxcar and (2) timing-active pixels had a peak rise time

within 0.07–1 s (in vivo, RCaMP1.07), 0.1–1 s (in vivo Lck-GCaMP6f) or 0.1–8 s (in vitro, Lck-GCaMP6f) compared to temporal boxcar. Active pixels were grouped within space (spatial radius of 4 μm) and time (0.2 s for RCaMP1.07 and 0.5 s Lck-GCaMP6f). The 3-D mask of active pixels was summed along the temporal dimension, normalized, and thresholded ($q = 0.2$) to make a 2D activity ROI mask. Raw image data from pixels within each 2D ROI were statistically compared to pixels surrounding the ROI ($p$-value < 0.05 by one-way ANOVA) to exclude false positives. Neuronal activity masks created by the algorithm and manually selected ROI mask were compared and the overlapping areas were excluded from the activity mask making sure each ROI was unique. For each ROI, a signal vector (dF/F) was calculated relative to the baseline fluorescence in the first 5 s of the trial, and signal events were detected using the findpeaks function in MATLAB. For each event, different features such as amplitude and peak onset latency were measured and finally exported as .csv files for statistical analysis. The peak onset latency was calculated from the smoothed signal trace (5 frame moving average) as the first time point when the signal went over the threshold of 2.5 times the standard deviation of the baseline after the start of stimulation. We categorized astrocytes Ca²⁺ events as fast and delayed based on the median onset latency of their respective neurons during stimulation (control; 1.71 s and Grin1 KD 1.78 s). For all analysis, activities within the stimulation window (0 < onset latency < 8 s) were compared to the equivalent in trials with no stimulation except for fraction of active pixels and repeated response score which compared the events during the whole trial. In order to compare data acquired with different magnifications (i.e., fields of view with different area sizes), the number of ROIs was normalized to the reported area of a mouse whisker barrel (32,000 μm²)[73]. For neurons, the amplitude distribution was described by a log-normal distribution ($R^2 > 0.95$). We categorized neurons based on this stable distribution using the 98th and 88th percentile of control neurons amplitude: High-responding neurons (amplitude ≥ 3.47 dF/F), Mid-responding neurons (1.92 ≤ amplitude < 3.47 dF/F), and low-responding neurons (amplitude < 1.92 dF/F). We also used a seed-based correlation analysis to correlate the signal vector (dF/F) for each ROI with the vectors from all other ROIs in the same field of view and examined the mean Pearson's correlation coefficient across each trial within stimulation window (0–8 s).

### Behavior tasks

The whisker touch-based novel texture recognition and novel object recognition tests were performed to assess whisker discrimination or recognition memory[3,47]. Both tasks included three general phases: habituation, learning, and testing. During the habituation phase, animals were introduced to an empty arena (40 cm × 40 cm) for 10 min a day for two consecutive days. The animal's performance inside the arena was monitored and animals with stress signs (e.g., no exploration or moving around) were excluded. For the novel texture recognition test, following the last session of habituation, animals were lightly anesthetized using isoflurane (4% induction, 1–1.5% maintenance), and all the whiskers on the whisker plate ipsilateral to the chronic window were trimmed back to the face using fine-tipped scissors. On the contralateral side of the nose, all whiskers except for those corresponding to the virus injection area were also trimmed. During the learning phase of the novel texture recognition test, animals were introduced to the arena with two stands (4 cm × 15 cm) covered with the same grade of sandpaper, e.g., 150 grit, placed in the middle of the arena with an equal distance from each other and walls. Animals were allowed to explore for 5 min. After the learning phase, animals were transferred to a resting cage for 5 min. During this time, the stands with sandpaper were removed from the arena, and two new stands were replaced: one familiar sandpaper (exactly the same as the one in learning phase) and a novel sandpaper (e.g., 220 grit). During the testing phase, animals were allowed to explore the arena with one

familiar and one novel texture for 3 min. The animal's performance was recorded using a camera. To minimize the impact of olfactory cues, three copies of each stand were used. Also, the arena and the objects were cleaned with 70% ethanol between the learning and testing phase and between animals. The novel object recognition test was performed in the same manner, where preference for a novel vs. familiar object was determined but with visually distinct objects. In this case, whiskers were not trimmed and objects of different color and size were used. For example, while in the learning phase, two 3-D printed blue bulldog shapes were used, but in the testing phase, one bulldog was replaced with an orange mouse shape.

### Behavior task analysis

Videos were analyzed either manually or using a deep learning software, DeepLabCut (DLC[74]), which allows for pose estimation of user-defined body parts using deep learning. DLC was first trained with 10 short clips (30 s) of the learning phase. The trained network was then used to estimate the position of animal's nose as well as the objects in each frame of the video. Finally, a .csv file was created including the position of marked targets across videos. In either case, the amount of time animals spent around each object in each phase was measured. Investigation time was defined as the total time the animal was facing towards the sandpaper/object and their nose within 2 cm of the object. Climbing over the objects or stands was not considered as investigation. Any animal that did not explore any objects in the learning phase, explored only one object in the testing phase, or had a total investigation time of less than 2 s in the learning or testing phase was excluded from the analysis. The discrimination index was defined as the difference between time spent exploring novel texture vs. familiar texture divided by total exploration time (Novel exploration time-familiar exploration time/total exploration time). While a discrimination index close to 1 implied a preference for the novel texture/object, a discrimination index close to 0 implied no preference.

### Fluorescence-activated cell sorting (FACS)

Animals were anesthetized by intraperitoneal injection of pentobarbital (150 mg/kg) and were transcardially perfused with $Mg^{2+}$- and $Ca^{2+}$-containing Hanks Buffered Salt Solution (HBSS$^{+Ca+Mg}$; NaCl, 140 mM; KCl, 5 mM; MgCl$_2$*6H$_2$O, 5 mM; MgSO4-7H$_2$O, 0.4 mM; CaCl$_2$, 1 mM; Na$_2$HPO4-2H$_2$O, 0.3 mM; NaHCO$_3$, 4 mM; KH$_2$PO4, 0.4 mM; D-Glucose, 6 mM). The injection area under the cranial window was cut into small pieces and was incubated with Dispase II (final concentration of 0.6U/ml; Millipore-Sigma; D4693) incubated at 37 °C for 1 h with gentle shaking (120 rpm). Then, the tissue was gently dissociated by passing through a 1 ml pipette tip 7 times, followed by a 40 μm pipette tip cell strainer (SP Bel-Art Flowmi). Cells were then spun at $400 \times g$ at 4 °C for 5 min. Cells were washed twice with ice-cold HBSS$^{+Ca+Mg}$ at $400 \times g$ for 5 min removing supernatant between each wash and keeping the pellet untouched. Cells were gently resuspended in ice-cold HBSS$^{+Ca+Mg}$ containing DNaseI (5U; Fisher Scientific; RQ1) and placed on ice until FACS. Single-cell sorting using a FACS machine (BD FACSAriaIII) was conducted by technicians at the Flow Cytometry core facility at the University of Manitoba. Astrocyte Lck-GCaMP6f fluorescence (i.e., green GFP fluorescence) was used for sorting. FACSDiva (Version 6.1.3) software was used for analysis of the recorded FACS events. For each signal, the side scatter was plotted against the fluorescence intensity.

### Quantification of *Grin1* expression using qPCR

Upon sorting GCaMP6f-expressing astrocytes, cells were lysed using the RNA Purification kit (ThermoFisher) according to manufacturer's instructions and were stored at −80 °C until RNA extraction. RNA was extracted using the Purelink RNA Micro Kit (ThermoFisher) according to manufacturer's instructions. RNA amplification was done using

MessageAmp II aRNA Amplification Kit (ThermoFisher) according to manufacturer's instructions except for the aRNA purification where we used Trizol to increase the recovery rate. Reverse transcription of aRNA was done according to SuperScript IV VILO (ThermoFisher). We used Qiagen PCR Purification kit for cDNA purification. Real-time PCR samples were prepared by using the purified cDNA (5 μL; 2 ng/μL), PowerUp SYBR MasterMix (Life Technologies; A25742) (10 μl), and primers (1 μl F and 1 μl R) which were all designed using NCBI primer blast software. PCR was done in triplicate on the QuantStudio 6 Flex with the following program: 2 min at 50 °C, 2 min at 95 °C, 45 cycles of 5 s at 95 °C, and 30 s at 60 °C. Analysis was done in QuantStudio 6 Flex software. All samples were gender and age matched (age of injection and days post injection, see Table 1). Using Normfinder and Bestkeeper software, 3 of the housekeeping genes found to be the most stable were used in the final analysis including *Actb*, *Atp5pb*, and *Hprt*. Relative expression of *Grin1* gene in Grin1 KD astrocytes was calculated based on previously generated standard curves (8-point dilution series), normalized to the expression of the 3 housekeeping genes, and finally compared to the average non-silencing relative gene expression. Primers sequences are as follows:

MusHPRT F667: 5′-ACAGGCCAGACTTTGTTGGA-3′; MusHPRT R765: 5′-CACAAACGTGATTCAAATCCCTGA-3′; MusACTB F902: 5′-TCCTTCTTGGGTATGGAATCCTG-3′; MusACTB R987: 5′-AGGTCTTTACGGATGTCAACG-3′; Atp5pb F: 5′-GTCCAGGGGTATTACAGGCAA-3′; Atp5pb R: 5′-TCAGGAATCAGCCCAAGACG-3′; Ywhaz F: 5′-ATCCCCAATGCTTCGCAACC-3′; Ywhaz R: 5′-ACTGGTCCACAATTCCTTTCTTG-3′; MusGrin1 1237 F: 5′-CAACATCTGGAAGACAGGACC-3′, MusGrin1 1308 R: 5′-CCAGTCACTCCATCTGCATAC-3′; Tubb3 F: 5′-ACCATGGACAGTGTTCGGTC-3′; Tubb3 R: 5′-AGCACCACTCTGACCAAAGATA-3′; S100b F: 5′-CTTCCTGCTCCTTGATTTCCTCCA-3′; S100b R: 5′-CGAGAGGGTGACAAGCACAAG-3′.

### RNA-seq analysis

Amplified RNA samples used for qPCR were sent for RNA sequencing at the University of British Columbia Sequencing and Bioinformatics Consortium. The concentration and quality of samples was confirmed on an Agilent Bioanalyzer and libraries were prepared with the Illumina Stranded mRNA kit (omitting the polyA tail pull-down step because of the amplified RNA). Amplified RNA was diluted to 5 μl with water. It was mixed with 15 μl of fragmentation master mix and denatured at 94 °C for 8 min. Following this, there was first and second-strand synthesis, adapter ligation, and PCR amplification. All samples were multiplexed, and sequencing was performed on an Illumina NextSeq 500 High Output. Upon sequencing, paired-end 74 bp reads were generated. Sequence reads were aligned to the mouse reference genome mm10 using HISAT2. The UCSC transcript data (mm10.ensGene.gtf) was then used to build feature count tables. In total there were 43,431 separate transcripts interrogated with 210,757,811 reads. Each *Grin1* sample averaged 9,920,465 reads while the controls averaged 15,701,617 reads per sample. These reads were fed into DESeq2 for differential expression analysis[75] with a false discovery rate (FDR) threshold <0.05. Two samples from the Grin1 KD group and 1 sample from the control group were excluded from the differential gene analysis because (a) these samples came out differently than other samples in their group during a PCA analysis of the RNA sequencing results, (b) they did not perform as expected during qPCR (higher or lower *Grin1* expression than other samples in the same group) which could suggest variability from previous preparation steps (FACS, etc.), and (c) there was very little RNA remaining after qPCR to be sent for sequencing, resulting in fewer sequencing reads for these samples. In the end, 7 Grin1 KD and 9 control samples were included in the analysis.

### Glial cultures and immunocytochemistry

Primary mixed glial cultures were prepared from cortices of newborn (up to P2) male CD1 mice. The cells isolated from cortices were plated

into T75 (75 cm²) flasks in minimal essential media (MEM, Thermo-Fisher; Cat # 11090099) supplemented with 2 mM GlutaMAX (Ther-moFisher; Cat #35050-061), 10% fetal bovine serum containing <0.05% endotoxin levels (ThermoFisher; Cat #12483-020) and 0.01% Strepto-mycin (ThermoFisher, Cat # 11860038), and maintained in 5% $CO_2$ at 37 °C in a humidified incubator. Media changes for mixed glia cultures were done at 1 and 4 days in vitro (DIV). Cells reached confluency ~8 DIV and consisted of astrocytes and microglia. On 15 DIV, the visible microglia were removed by shaking and remaining mixed glial cells were detached with trypsinization and treated with neuraminidase (100 mU/ml; Sigma–Aldrich; N7885) in culture media for 2 h. Cells were washed 3 times (centrifuged at $100 \times g$ and resuspended in media) to remove the neuraminidase. Then, they were plated in 6 well plates with glass coverslips at $2 \times 10^6$ cells/well and AAV virus was added: Grin1 KD (AAV9-sGFAP-Grin1-shRNA$^{mir}$-Lck-GCaMP6f) or con-trol (AAV9-sGFAP-NS-shRNA$^{mir}$-Lck-GCaMP6f) at a multiplicity of infection of 20,000 particles/cell.

Two weeks after viral transduction, cells on the coverslips were fixed with 1% paraformaldehyde for 10 min. Coverslips were washed three times with TBST (Tris-buffered saline + Tween 20; 0.05%) for 10 min. Then, they were incubated in blocking solution including TBS + 0.3% Triton X-100 (0.3% TBST) + 5% Normal Donkey Serum (NDS) + Donkey anti-mouse IgG Fab fragment (90 μL/ml; for mouse-on-mouse blocking; Jackson Immunoresearch; 715-007-003) for 30 min. After washing 3 times with 0.05% TBST for 5 min, the cells were incubated in 0.3% TBST + 2.5% NDS + primary antibodies for 1 h at 4 °C. Following 3 more washes with TBST (0.05%) for 10 min, secondary antibodies were added in 0.05% TBST and 1% NDS for 30 min. Cells were then washed twice with TBST (0.05%) and once with TBS before the coverslips were mounted on glass slides. Images were acquired on a confocal microscope (Zeiss LSM 810) at 20× magnification. Primary antibodies included Chicken-anti-GFP (1:1000, Aves; GFP 1020), Rabbit-anti-GFAP (1:3000, Dako; Z0334), and Mouse-anti-GluN1 (1:200, Millipore; MAB363). Secondary antibodies include Donkey-anti-Chicken-IgYY-488 (1:1000, Invitrogen; A78948), Donkey-anti-Mouse-IgG-647 (1:1000, Invitrogen; A31571), and Donkey-anti-Rabbit-IgG-568 (1:1000, Invitrogen; A10042).

### Immunohistochemistry

Animals were anesthetized by intraperitoneal injection of pento-barbital (150 mg/kg) and were rapidly transcardially perfused with 20 mL ice-cold oxygenated aCSF followed by 60 ml ice-cold 2% par-aformaldehyde (PFA; Millipore-Sigma; P6148) (in 2× Phosphate-buffered saline (PBS), pH adjusted to 7.2–7.4). Hemispheres were then post-fixed in 4% PFA for 3 h, washed with ice-cold 1× PBS, and cryoprotected with 30% sucrose (in 1× PBS) at 4 °C overnight. Tissue frozen in Optimal cutting temperature compound (OCT compound; Tissue Tek) was cut on a cryostat (Leica) to obtain 40 μm thick sec-tions. The sections selected for staining were rinsed in 1× PBS and then were washed twice with TBS + 0.3% Triton X-100 (0.3% TBST) for 10 min while incubated on a rocking platform at room temperature. Sections were then incubated in blocking solution including 0.3% TBST + 5% Normal Donkey Serum (NDS) + Donkey anti-mouse IgG Fab fragment (90 uL/ml; Jackson Immunoresearch; 715-007-003) for 1 h. Subsequently, sections were washed 3 times with 0.3% TBST for 5 min. Sections were incubated in 0.3% TBST + 2.5% NDS + primary antibodies overnight at 4 °C on a rocking platform. Following incubation with primary antibodies, sections were washed with 0.05% TBST 3 times for 10 min and were incubated with secondary antibodies and 0.05% TBST + 1% NDS for 1 h at room temperature. Following incubation, sections were washed twice with 0.05% TBST and once with TBS for 10 min. Sections were then washed in 0.1× PBS and mounted on glass slides with a coverslip. Z-stack images of sections were acquired on a confocal microscope (Zeiss LSM 810) at 63× magnification. Primary antibodies included Chicken-anti-GFP (1:1000, Aves; GFP 1020),

Rabbit-anti-GFAP (1:3000, Dako; Z0334), and Mouse-anti-GluN1 (1:200, Millipore; MAB363) or Mouse anti-NeuN (1:200, Millipore; MAB377). Secondary antibodies included Donkey-anti-Chicken-IgY-488 (1:1000, Invitrogen; A78948), Donkey-anti-Mouse-IgG-647 (1:1000, Invitrogen; A31571), and Donkey-anti-Rabbit-IgG-568 (1:1000, Invitrogen; A10042).

### Immunocytochemistry and immunohistochemistry image quantification

Image J was used to analyze the intensity of GluN1 signal to compare it between control and Grin 1 KD transduced cultured astrocytes and astrocytes in tissue sections. In each field of view, 3–4 astrocytes were identified using the GFAP and GFP staining. For each of the astrocytes, a mask was drawn and saved using the GFP channel (for cultured cells) and GFAP channel (for in situ astrocytes). The same mask was subsequently applied on the GLUN1 channel and the inte-grated density as well as the area of the astrocyte was noted. Finally, background intensity of the GLUN1 channel was also recorded. Corrected Total Cell Fluorescence (CTCF) was calculated using the following formula:

CTCF = Integrated Density – (Area of astrocyte X Background mean fluorescence)

To calculate the GLUN1 intensity differences in neuron GLUN1 depleted vs neuron non-GLUN1 depleted areas from tissue sections, overall field intensity in control field of views were compared and a Corrected Total Field Fluorescence (CTFF) was calculated as follows:

CTFF = Integrated Density – (Overall field area X Background mean fluorescence)

### Statistics & reproducibility

All statistics were performed in R (Version 1.2.1335). Metrics with a nested design were analyzed using the lme4 package for linear mixed-effects models. For example, with in vivo $Ca^{2+}$ imaging, multiple ROIs were identified per field of view per trial, and multiple fields of view were imaged per animal across several imaging sessions. Also, with immunocytochemistry and immunohistochemistry multiple astro-cytes were analyzed per culture or brain slice. Therefore, linear mixed models are needed to appropriately account for related effects (known as random effects) when multiple measurements are from the same sample (i.e., each measurement is not an independent n). For in vivo $Ca^{2+}$ analysis, when generating the linear models we included random effects with intercepts for individual animals, fields of view, ROIs, and trials. Then we added our fixed effects to be tested, which included animal type (control or Grin1 KD), stimulus condition (with or without stimulation), astrocyte type (fast or delayed), neuronal populations (High or mid or low-responding neurons), and sex (male or female) as well as the interaction of these effects. Likelihood ratio tests compar-ing models with fixed effects against models without fixed effects were performed to determine the model with the best fit while accounting for the different degrees of freedom. All data were reported and plotted as uncorrected means and standard error of the means. P-values for different parameter comparisons were obtained using the multcomp package with Tukey post hoc tests. Non-parametric Mann–Whitney–Wilcoxon tests were performed in R (Version 1.2.1335) for qPCR and behavior tasks.

Though representative images of immunocytochemistry and immunohistochemistry experiments are shown in Fig. 1, the staining was reproduced multiple times. In case of immunocytochemistry, three replicates of glial cultures and viral transduction were performed in both control and Grin1 KD groups. From each culture, 3–5 fields of view were imaged and 5 cells from each field of view were quantified. In case of immunohistochemistry, 3 control and 3 Grin1 KD mice were used, and again, 5 fields of view and 3–5 cells per field of view were quantified. Representative two-photon images in Figs 2, 3, and 4 were selected as examples for all slices or in vivo fields of view represented in the graphs in those figures.

## Reporting summary

Further information on research design is available in the Nature Portfolio Reporting Summary linked to this article.

## Data availability

Requests for further information, reagents, data, and resources should be directed to and will be fulfilled by Dr. Jill Stobart (jillian.stobart@umanitoba.ca). Due to the large size of the imaging data collected during this project, files are currently stored on our University of Manitoba servers, but will be made available on request. AAVs for astrocyte Grin1 KD and control are available for order at the University of Zurich Viral Vector Facility. The RNA-seq data has been uploaded to NCBI SRA (BioProject ID PRJNA1063450). Source data are provided as a Source Data file.

## Code availability

The code used for calcium analysis (CHIPS v1.1.0) is available as a MATLAB toolbox at (//github.com/EIN-lab/CHIPS/releases; RRID:SCR_015741)[36]. Requests for further information should be directed to Dr. Jill Stobart (jillian.stobart@umanitoba.ca).

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

## Acknowledgements

We would like to thank the staff in the Central Animal Care Services and Veterinary Services at the University of Manitoba for their diligent work and dedication to animal care and husbandry. We would also like to thank (a) Christine Zhang and the Flow Cytometry Core Platform at the University of Manitoba for assistance with cell sorting and (b) the University of British Columbia Sequencing and Bioinformatics Consortium and Richard Leduc and the Children's Hospital Research Institute of Manitoba for assistance with RNA sequencing and analysis. Images presented in this publication were collected in the Live-Cell Imaging Facility at the University of Manitoba. This work was supported by a Discovery Grant (RGPIN-2020-088, J.S.) from the Natural Sciences and Engineering Research Council of Canada (NSERC), the University

Research Grants Program (University of Manitoba), and start-up funding from the University of Manitoba. The Stobart lab has also received support from the Canadian Institutes for Health Research, Research Manitoba, Brain Canada/Azrieli Foundation, and the Manitoba Medical Service Foundation. M.K., J.M.R, and S.S. were supported by graduate studentships from the Rady Faculty of Health Sciences and Research Manitoba. T.S. and J.P.N. were supported by Mitacs Globalink Internships. D.E. was supported by an Undergraduate Research Award (University of Manitoba).

## Author contributions

M.S. and J.S. designed the study. N.A., M.K., J.M.R., S.S., J.P.N., M.S., A.M., T.S., F.O., D.E., S.C.-F. performed the experiments. B.D.G. created equipment needed for the experiments. A.L. prepared the astrocyte-enriched cultures. N.L. and T.M.K. provided resources and helped with the immunocytochemistry experiments. B.W. and M.F.J. provided resources and critical feedback. N.A., M.K., M.S., and J.S. analyzed data. N.A., M.K., and J.S. wrote the paper.

## Competing interests

The authors declare no competing interest.
