## [Peer Review File · Nature Communications]

Cortical astrocyte N-Methyl-D-Aspartate receptors influence whisker barrel activity and sensory discrimination in miceREVIEWER COMMENTS

Reviewer #1 (Remarks to the Author):

The paper by Ahmadpour explores the physiological role for glial NMDA receptors in the astrocyte function and astrocyte-driven modulation of neuronal signalling and behaviour. Authors used a novel knockdown approach to selectively reduce astrocyte N-methyl-D-aspartate receptors, based on the miRNA-adapted shRNA approach to knockdown (KD) GluN1 NMDAR subunit selectively in the astrocytes of the barrel cortex. This approach was combined with the state-of-the-art in vivo 2-photon fluorescent recordings of astrocytic and neuronal activity and behavioural experiments. The main strength of the paper is clear and convincing demonstration of significant changes in phenotypes of astroglial Ca²⁺-signalling, neuronal synchronisation and adaptation and sensory discrimination in the astrocytic-Grin1 knock-down mice. The reported results provide new insights into physiological relevance of astroglial NMDA receptors and mechanisms of astrocyte-neuron communications as a whole.

The paper seems to be suitable for publication in the Nature Communications, the presented results will surely attract a great interest among wide readership of the journal. Yet, there are several issues which need to be addressed to enhance the message and substantiate the conclusions of the work.

1. The only serious downside of the paper is a gap in its narrative between founding of the deficit in the astrocytic Ca²⁺-signalling in cortex of Grin1 KD mice and changes in neuronal activity and behaviour. The putative mechanism which could underlie a feedback between the activation of astrocytic NMDARs and activity of neurons, in particular synchronization and adaptation, remains unclear, which make two main parts of results somewhat disjointed. To bridge this gap, some additional experiments are required.

In the Discussion, Authors suggest that effect of deficiency in astrocytic Grin1 and Ca²⁺-signalling may be related to the impairment of ATP release from astrocytes, which seems to be a viable and attractive hypothesis, since there are several mechanisms by which astrocyte-derived ATP can mediated astrocyte-driven widespread modulation of local synaptic networks.

First, astrocyte-derived ATP can down-regulate both the phasic and tonic inhibition; Authors rightfully recognised this in their interesting notion of "inhibition blanket" (page 14, 1st paragraph). Second, the astrocyte-derived ATP could affect synaptic signalling and neuronal synchronisation via pre-synaptic P2 or A1 receptors. The weak side of "ATP gliotransmission" for explanation of the Grin1 KD effect is that NMDARs are not an exclusive way of elevation of cytosolic Ca²⁺ in astrocytes. For instance, release of ATP from astrocytes can be triggered via IP3 pathway by CB1, mGluR and α 1-adrenoreceptors.

Alternative hypothesis, which is much more "NMDAR-centric" is a dominant role of NMDARs in the Na⁺, rather than Ca²⁺-signalling. The contribution of NMDARs cannot be rivalled by the astrocytic AMPARs, due to their fast desensitization and much lower sensitivity to glutamate.

The NMDAR-mediate Na⁺-influx in astrocytes can down-regulate the astroglial GABA transport which in turn could affect neuronal firing.

Could Authors discriminate between these hypotheses by showing that either purinergic antagonists or GABA transporter inhibitors can produce the similar effect on neuronal synchronization and adaptation as Grin1 KD ? It would also be helpful to highlight the specific role for astrocytic NMDARs by showing that knock-down of astrocytic IP3R receptors did not lead to the same neuronal signalling and behavioural phenotypes as Grin1 KD.

Other (more technical) issues:

1) Analysis of astrocytic Ca-transients (Fig.2) : although the data presented in the Fig.2 strongly suggest the difference in the Ca²⁺-signalling in the Grn1KD vs control mice, it is not very informative. The only representatives of GCaMP6 fluorescence are wide-field, low-scale images in panels C,D which do not allow to associate bright spots (supposed microdomains of Ca²⁺-activity) with any astrocytes. The accompanying traces of GCaMP fluorescence shown look more like steady

elevation of baseline, rather than transient events which contrasts to the waveforms of astrocytic signalling showed in the Fig.3C. Complementing these images with the examples of high-scale images of astrocytes with identified microdomains and examples of individual events – with references to specific MD in the images – would be very helpful and convincing.

2) continuing with Fig.2: the Area-under-the-curve, which is used as a readout for Ca²⁺-signalling, is determined both by the magnitude and kinetics of the Ca²⁺-transients, showing some specific data on these parameters would be very helpful. In particular, what was the mean amplitude of transients in the Grin1 KD vs control and for how long did the NMDA-elicited responses last? What was the decay time, KD vs control?

The statistical diagrams which represent numerous ROIs piled together for all KD or control mice used, exhibit rather wide distribution which brings few more questions:

- i) does the spread distribution reflect an intrinsic variability among different ROI or it is related to the variation of astrocytic Ca²⁺-signalling between individual mice;
- ii) could Authors also present the diagram showing numbers of ROI peaks/per minute averaged for individual mice and then compare such data for KD and control mice
- ii) would the difference between KD and control be still statistically significant if average numbers of ROI peaks/per calculate for individual mice were compared?

3) As it is, the main focus point of the Fig3 – the panel C, showing the examples of Ca²⁺-signalling in astrocytes and neurons, is very confusing and lack consistency in few aspects.

(i) different pairs of astrocytes are shown for the baseline (no stim.) and stimulation conditions but they all marked as A1, A2 which is confusing. It is possible that different astrocytes exhibit spontaneous and stimulus-elicited activity but such heterogeneity deserves a deep attention so all four astrocytes should have been tracked (i.e.A1 – A4). Furthermore, more detailed statistical analysis of such “role-switching” in the KD vs control animals might give more insights into role for NMDARs in the astrocyte function;

(ii) examples, shown in the Fig.3C for Grin1 KD exhibit no marked spontaneous activity for both astrocyte 1 and 2 which contradicts the statement that knockdown of astrocyte NMDARs did not affect the number of spontaneous Ca²⁺ events in astrocytes (page 11, the last paragraph) and statistical diagram on the ROI amplitude in Fig.3F – Did panel Fig.3C show the proper, representative examples of ROI activity?

(iii) Fig.3F – the mean±SEM diagram (bar plot on the right) suggests that the mean amplitude of Ca²⁺-transients in astrocytic ROIs of Grin1 KD mice is larger than in their control counterparts; although there is no P value, judging by the size of error bars, the difference may be statistically significant. This result seems to contradict to other data depicted in the Fig.3. How do Authors explain this?

4) The figure 4, to a large extent, hangs in the air. Apparently, the data it shows have been obtained using the same experimental paradigm, but this should be at least mentioned in the legend. To enhance the message of Figure 4, and make it more coherent to other figures, showing some representative images and time courses of hSyn-RCaMP1 fluorescence would be very helpful. For instance, panel A mentions “spontaneous neurons and dendrites” but it is hard to visually discriminate any dendrites in the example of hSyn-RCaMP1 fluorescent showed in the Fig.3C. So, showing high-scale representative images for Grn1 KD and control is needed. Similarly, the panels B and C mention the amplitude of spontaneous and stimulation evoked ROI signalling, whereas panel C shows the numbers of responsive vs. non-responsive neurons. These pooled data should be accompanied by the representative examples of base-line and stimulus evoked fluorescent signals both for responsive and non-responsive ROIs. Showing the response heatmaps for the KD and control neurons (as it was done for the astrocytes in the Fig.3) would also be beneficial.

5) In many figures, the statistical data are presented as violin plots and the statistical significance is said to be calculated using linear mixed models method. Although this method is suitable to verify the statistical difference between KD and control in the context of the work, a wider audience might not be familiar with details of this method, so more detailed explanation of linear

mixed models is in order. Also, some explanation of the meaning of X and Y parameters in violin plot would be helpful, either in the legend to some figure (e.g. Fig2) or in Methods.

Reviewer #2 (Remarks to the Author):

In their manuscript "Cortical astrocyte N-Methyl-D-Aspartate receptors influence whisker barrel activity and sensory discrimination" Ahmadpour et al. describe the physiological and behavioral effects of knocking down NMDAR expression in astrocytes, achieved by shRNA hairpins for knockdown of Grin1 mRNA (Grin1 KD). The construct, that also have Lck-GCaMP is delivered by AAV9, reduces the expression of Grin1 in astrocytes by 70%, and decreases NMDAR activity in acute adult brain slices. They further show Grin1KD diminished sensory-evoked calcium events in vivo in astrocytes in the barrel cortex, and different neuronal responses (diminished responses but with higher amplitude) to stimulation in vivo. Then they tested the levels of synchronicity during stimulation: they found that controls had lower BL synchronicity which increased during stimulation, and Grin1KD had higher synchronicity without stimulation, which decreased during stimulation. They then analyze the delayed response from the time of stimulation to the beginning of the Ca²⁺ response, which was the same in neurons, and reduced in astrocytes, injected with the KD virus. Finally, they checked behavior and found that Grin1KD have a problem with detecting small (but not large) grit differences.

I am not impressed by the study, as it has a very preliminary feel. The impression is of a list, rather than a layered work. In the bottom line there is a single 2-photon study (even if it is given in 4 figures), and an additional small behavioral study. This work will require more experiments, and better analyses of the existing one, before I can recommend publication.

Major:

- 1) I would like to see imaging and behavior joined - there are a lot of studies with head fixed behavior under a 2-photon in the barrel Cx. And then you can really say something about the connection between them.
- 2) In addition, I would have liked to see at least a beginning of a mechanism. How do the astrocytes affect neurons? Why in some cases there's a big difference in the astro, but not in the neurons (Fig 6)? Etc...
- 3) You analyze the astro and neurons separately. Why? Is there nothing to be extracted from the fact that you do double imaging?
- 4) The effect on neurons amplitude (Fig 4) is very small. $P < 0.01$ for more than thousand ROIs per group, is not that impressive (Fig 4D). Then you divide it by responsiveness, and find changes in ROIs/FOV only in low responsive cells. I understand that the authors give an explanation " Low and mid-responsive neurons are more weakly recruited during repeated trials of sensory stimulation, but they have a greater capacity for reorganization during changes in sensory input". But can they preclude the alternative explanation that this group is comprised from most of the neurons (88%) so it is easier to find significance, but not in the smaller groups?
- 5) It is very difficult to understand the figures. For example Fig 3D, Fig4B,D - what's in the grey part? Is it all the data (i.e. ROIs) or the data per mouse? Fig 4 (as another example): Dark purple color is used to mark 2 different things (F and the rest), and Grin1KD is light purple, except than in E.
- 6) Fig 6 - is the astrocyte are changed, with no effect on the neurons - what does it mean??

Minor

- 1) The Abstract doesn't do the paper a good service...
- 2) Figure 1 - it's really not necessary to show the controls in the main figure...
- 3) P 8, row 6 "...99th" do you mean 88th?
- 4) The discussion is VERY long, and have little that was not said before.

Reviewer #3 (Remarks to the Author):

The functional characterization of neuronal NMDA receptors has been awarded a Nobel Prize and has become textbook knowledge for biology students. Much less is known about NMDA receptors in glial cells. The work described in the manuscript by Noushin Ahmadpour and colleagues significantly adds to our current understanding of NMDA receptors in a major subtype of glial cells, astrocytes. The authors used a multi-shRNA AAV approach to knockdown the constitutive NMDA receptor subunit GriN1 in astrocytes and then employed a combination of whisker stimulation and in vivo two-photon calcium imaging to investigate the impact of NMDA receptor-mediated astroglial calcium signals, neuronal activity, and somatosensory behavior.

The main observations are:

1. AAV-shRNA reduces GriN1 mRNA levels by about 70% in FAC-sorted astrocytes.
2. In brain slices, fewer NMDA/glycine-evoked calcium signals are found compared to control preparations.
3. In awake animals, whisker stimulation cannot trigger higher amplitude calcium signals in GriN1 knockdown astrocytes.
4. The Ca²⁺ signals of neurons with high responsiveness to whisker stimulation are blocked in astroglial GriN1 knockdown barrel fields.
5. Whisker stimulation in control mice evoked more synchronous calcium signals in neurons than in those of astroglia-GriN1 deficient mice.
6. The numbers of stimulus-evoked calcium microdomains were reduced in GriN1-knockdown astrocytes versus controls.
7. In a somatosensory behavioral experiment (detection of sandpaper with differently sized grains by the whiskers) the mice with GriN1 deficiency were not able to distinguish the sandpapers. However, the sandpaper itself elicited neuronal and astroglial calcium responses.

From their observations the authors conclude that astrocytes detect distinct neuronal activity via their NMDA receptors. The astrocytes then enhance synchronization of adjacent neurons. The combined neuronal and astroglial communication is essential to distinguish fine differences in whisker stimulation.

This is a very interesting observation and, so far, the best experimental evidence how astroglial NMDA receptors contribute to a distinct brain function.

In general, the work is well described and performed, although some critical points remain:

1. The extent of astroglial GriN1 knockdown has only been shown by a qPCR of GriN1 from FACsorted astrocytes. The functional analysis only indirectly suggests that also the receptor channel complex is missing. Here, at least two of three types of additional experiments are required: 1. Whole-cell patch-clamp recordings of control and mutant astrocytes in slices to complement the calcium traces of fig. 2 C/D. In addition, the time scale should be magnified. 2. Immunohistochemistry of GriN1 on control and mutant cells. The cell-specific localization of GriN1 on astrocytes will be facilitated by the membranous Ick-GCaMp expression. 3. qPCR of the NMDA receptor subunits to check for compensatory mechanisms.

These additional experiments are necessary since about 50 % mutant and GriN1-KD astrocytes display the same level of GriN1 mRNA. The patch-clamp and/or the immunohistochemical analysis would indicate to which extent the functional receptor is degraded and removed from the membrane.

2. A more detailed localization of GriN1 should be deduced from a cellular map of microdomains along processes and at process tips, in particular to distinguish perisynaptic from endfeet signals. For that purpose, from control and mutant astrocytes the individual calcium microdomains should be detected and, for example, plotted as a localization heatmap onto the outline of schematic astrocyte.

3. Why were some complete sets of experiments (slice imaging, whisker discrimination) only performed with female mice?

Minor points:

Figure 6: The title of the legend is misleading. All type of microdomains are reduced, not only those with rapid onset. In A, the description of the X-axis is missing.

Response to Referees

We thank the reviewers for their insightful comments and suggestions. The revision process has helped to significantly improve our manuscript in several ways. We have added new experiments to strengthen the evidence supporting our Grin1 knockdown approach and to provide mechanistic insight linking astrocyte NMDA receptors with cortical neuron activity and sensory information processing through purinergic signaling. We also have generated new figures to better convey our findings. All new text within the manuscript is highlighted in red. Below please find the reviewer point-by-point comments (in black text) and our response (in blue text).

Reviewer #1 (Remarks to the Author):

The paper by Ahmadpour explores the physiological role for glial NMDA receptors in the astrocyte function and astrocyte-driven modulation of neuronal signalling and behaviour. Authors used a novel knockdown approach to selectively reduce astrocyte N-methyl-D-aspartate receptors, based on the miRNA-adapted shRNA approach to knockdown (KD) GluN1 NMDAR subunit selectively in the astrocytes of the barrel cortex. This approach was combined with the state-of-the-art in vivo 2-photon fluorescent recordings of astrocytic and neuronal activity and behavioural experiments.

The main strength of the paper is clear and convincing demonstration of significant changes in phenotypes of astroglial Ca²⁺-signalling, neuronal synchronisation and adaptation and sensory discrimination in the astrocytic-Grin1 knock-down mice. The reported results provide new insights into physiological relevance of astroglial NMDA receptors and mechanisms of astrocyte-neuron communications as a whole.

The paper seems to be suitable for publication in the Nature Communications, the presented results will surely attract a great interest among wide readership of the journal. Yet, there are several issues which need to be addressed to enhance the message and substantiate the conclusions of the work.

1. The only serious downside of the paper is a gap in its narrative between founding of the deficit in the astrocytic Ca²⁺-signalling in cortex of Grin1 KD mice and changes in neuronal activity and behaviour. The putative mechanism which could underlie a feedback between the activation of astrocytic NMDARs and activity of neurons, in particular synchronization and adaptation, remains unclear, which make two main parts of results somewhat disjointed. To bridge this gap, some additional experiments are required.

In the Discussion, Authors suggest that effect of deficiency in astrocytic Grin1 and Ca²⁺-signalling may be related to the impairment of ATP release from astrocytes, which seems to be a viable and attractive hypothesis, since there are several mechanisms by which astrocyte-derived ATP can mediated astrocyte-driven widespread modulation of local synaptic networks.

First, astrocyte-derived ATP can down-regulate both the phasic and tonic inhibition; Authors rightfully recognised this in their interesting notion of “inhibition blanket” (page 14, 1st paragraph). Second, the astrocyte-derived ATP could affect synaptic signalling and neuronal synchronisation via pre-synaptic P2 or A1 receptors. The weak side of “ATP gliotransmission” for explanation of the Grin1 KD effect is that NMDARs are not an exclusive way of elevation of cytosolic Ca²⁺ in astrocytes. For instance, release of ATP from astrocytes can be triggered via IP3 pathway by CB1, mGluR and α 1-adrenoreceptors.

Alternative hypothesis, which is much more “NMDAR-centric” is a dominant role of NMDARs in the Na⁺,

rather than Ca²⁺-signalling. The contribution of NMDARs cannot be rivalled by the astrocytic AMPARs, due to their fast desensitization and much lower sensitivity to glutamate.

The NMDAR-mediate Na⁺-influx in astrocytes can down-regulate the astroglial GABA transport which in turn could affect neuronal firing.

Could Authors discriminate between these hypotheses by showing that either purinergic antagonists or GABA transporter inhibitors can produce the similar effect on neuronal synchronization and adaptation as Grn1 KD ? It would also be helpful to highlight the specific role for astrocytic NMDARs by showing that knock-down of astrocytic IP3R receptors did not lead to the same neuronal signalling and behavioural phenotypes as Grn1 KD.

We thank the reviewer for these helpful comments and suggestions, and fully agree that mechanistic insight is needed to enhance this story. Therefore, we provide several new figures supporting: 1) that reduced purinergic signaling is a major contributor to altered neuronal activity and synchronization after knockdown of astrocyte NMDARs (Figure 8) and 2) that knockout of IP3R2 does not replicate changes we observe after astrocyte NMDAR KD (Supplementary Figure 5).

First, we found we could rescue the effects of Grn1 KD on neurons by administering ATPγS topically to the surface of the brain after the cranial window was removed. This improved the neuronal response to whisker stimulation, neuronal synchronization and increased the number of low responding neurons that are recruited during whisker stimulation. Animals also performed better during the whisker discrimination task. Together this suggests that a lack of purines contributes to neuronal impairment after reduced astrocyte NMDAR expression and we suggest this is due to decreased ATP gliotransmission. Future work clearly linking astrocyte NMDAR activity with purinergic release will be an exciting follow-up to the present manuscript. We acknowledge that Na⁺ signaling and its impacts on neurotransmitter uptake (particularly GABA) by astrocytes is also an attractive alternative hypothesis to mechanistic changes after astrocyte Grn1 KD. This warrants future investigation but was beyond the current study.

Previous studies have shown that IP3R2 knockout mice have reduced calcium signaling in astrocytes¹⁻⁴. Using the same neuronal calcium imaging paradigm as for Grn1 KD, we investigated neuronal responses to whisker stimulation in awake IP3R2 knockout mice. We found that in IP3R2KO compared to wildtype littermates: a) spontaneous neuronal activity was not elevated, b) an increased number of neurons responded to sensory stimulation, and c) neurons were synchronized during whisker stimulation (Supplementary Fig. 5). These results show no significant neuronal impairment, unlike neurons in Grn1 KD mice, suggesting the effects we observed in Fig. 4 were specific to a reduction in astrocyte NMDARs and not only due to a reduction in astrocyte Ca²⁺ events. Since we did not observe any neuronal impairments in awake IP3R2 knockout mice, we did not proceed with the novel texture recognition behavior task.

Other (more technical) issues:

1) Analysis of astrocytic Ca-transients (Fig.2) : although the data presented in the Fig.2 strongly suggest the difference in the Ca²⁺-signalling in the Grn1KD vs control mice, it is not very informative. The only representatives of GCaMP6 fluorescence are wide-field, low-scale images in panels C,D which do not allow to associate bright spots (supposed microdomains of Ca²⁺-activity) with any astrocytes. The

accompanying traces of GCaMP fluorescence shown look more like steady elevation of baseline, rather than transient events which contrasts to the waveforms of astrocytic signalling showed in the Fig.3C. Complementing these images with the examples of high-scale images of astrocytes with identified microdomains and examples of individual events – with references to specific MD in the images – would be very helpful and convincing.

We have now provided different example data and traces in Fig. 2. Updated panels B-D are meant to provide clearer examples of control and Grin1 KD fields of view where all ROIs are marked with green or white outlines. Typically, we see observe ROIs with elevated calcium dynamics during the agonist application (example traces; Fig. 2B, C). Bath application of the agonists commonly resulted in higher amplitude peaks in control ROIs, which caused a more sustained response in the average of all control ROIs (Fig. 2D, black line).

2) continuing with Fig.2: the Area-under-the-curve, which is used as a readout for Ca²⁺-signalling, is determined both by the magnitude and kinetics of the Ca²⁺-transients, showing some specific data on these parameters would be very helpful. In particular, what was the mean amplitude of transients in the Grin1 KD vs control and for how long did the NMDA-elicited responses last? What was the decay time, KD vs control?

We have now plotted the mean amplitude in Fig. 2 to better illustrate how this compares between control and Grin1 KD slices. We did not find a difference in Ca²⁺ event duration between control and Grin1 KD (data not shown, but mentioned in the manuscript text), suggesting that amplitude was the main contributor to increased AUC.

The statistical diagrams which represent numerous ROIs piled together for all KD or control mice used, exhibit rather wide distribution which brings few more questions:

- i) does the spread distribution reflect an intrinsic variability among different ROI or it is related to the variation of astrocytic Ca²⁺-signalling between individual mice;
- ii) could Authors also present the diagram showing numbers of ROI peaks/per minute averaged for individual mice and then compare such data for KD and control mice
- ii) would the difference between KD and control be still statistically significant if average numbers of ROI peaks/per calculate for individual mice were compared?

To make the data and possible variability clearer, we have now plotted all data for individual ROIs as violin plots (left side graphs) and the mean data for each animal (i.e. each brain slice) as a bar graph with dots to represent individuals (right side graphs). Please see the updated plots in Fig. 2E-G. Statistics were calculated using linear mixed models for the violin plots to account for the nested study designed (i.e. ROIs were collected from the same slice and were therefore not unique individuals), while bar graphs stats were calculated using Kruskal-Wallis tests and paired Wilcoxon tests with Bonferroni correction (for p values) because each n was an average for that slice.

3) As it is, the main focus point of the Fig3 – the panel C, showing the examples of Ca²⁺-signalling in astrocytes and neurons, is very confusing and lack consistency in few aspects.

(i) different pairs of astrocytes are shown for the baseline (no stim.) and stimulation conditions but they all marked as A1, A2 which is confusing. It is possible that different astrocytes exhibit spontaneous and stimulus-elicited activity but such heterogeneity deserves a deep attention so all four astrocytes should have been tracked (i.e. A1 – A4). Furthermore, more detailed statistical analysis of such “role-switching” in the KD vs control animals might give more insights into role for NMDARs in the astrocyte function;

We would like to note that for each imaging trial (with and without whisker stimulation) we analyzed the movies to extract regions of interest that were “active” (i.e. had calcium events) in that individual movie. As such, different astrocyte ROIs were identified in each trial. Heat maps of the activity were generated to show areas that were active in multiple trials (see Fig. 3E for example), but classical “role-switching” analysis is challenging because the same microdomain regions were not necessarily captured in trials with and without stimulation.

We now provide example maps of ROIs identified based on their activity in trials with and without stimulation (Fig 3E, F) to make it clear that a similar level of spontaneous activity occurs in control and Grin1 KD, but there is less microdomain recruitment during stimulation in Grin1 KD. We also overlaid all the active ROIs from the same field of view across multiple trials to emphasize that more areas are activated repeatedly with whisker stimulation in control astrocytes. Finally, we also now provide calcium movies of control astrocytes (Supplementary Movie 1) and Grin1 KD astrocytes (Supplementary Movie 2) that illustrate how fewer astrocyte Ca²⁺ events occur during whisker stimulation in KD.

(ii) examples, shown in the Fig.3C for Grin1 KD exhibit no marked spontaneous activity for both astrocyte 1 and 2 which contradicts the statement that knockdown of astrocyte NMDARs did not affect the number of spontaneous Ca²⁺ events in astrocytes (page 11, the last paragraph) and statistical diagram on the ROI amplitude in Fig.3F – Did panel Fig.3C show the proper, representative examples of ROI activity ?

We thank the reviewer for this feedback and agree that the previous example traces were perhaps not illustrative of the further effects of Grin1 KD that we described in the rest of the figure. We have provided new representative examples traces in Figure 3K and L. These traces are from different astrocyte compartments, somata and endfeet that were manually selected based on the SR101 signal and processes that were selected by the activity-based algorithm. Both Grin1 KD and control show similar traces during trials without stimulation. However, during stimulation, control MDs are more aligned with the stimulus and the amplitude is increased. There is also no obvious difference in Grin1 KD Ca²⁺ amplitude between the traces with stimulation and without.

Similar dynamics are also apparent in the new Supplementary Movies we have provided.

(iii) Fig.3F – the mean±SEM diagram (bar plot on the right) suggests that the mean amplitude of Ca²⁺ - transients in astrocytic ROIs of Grin1 KD mice is larger than in their control counterparts; although there is no P value, judging by the size of error bars, the difference may be statistically significant. This result seems to contradict to other data depicted in the Fig.3. How do Authors explain this ?

We also noted the spread of the data from Fig. 3F (now Fig. 3H) suggesting astrocyte calcium amplitude increases in Grin1 KD particularly when considering the violin plot during no stim trials. However, this was not different than controls within the statistical model ($P=0.7$). It is tempting to speculate that elevated spontaneous neuronal activity in Grin1 KD (Fig. 4) could increase amplitude of astrocyte Ca^{2+} microdomains, but statistically, this was not the case.

4) The figure 4, to a large extent, hangs in the air. Apparently, the data it shows have been obtained using the same experimental paradigm, but this should be at least mentioned in the legend. To enhance the message of Figure 4, and make it more coherent to other figures, showing some representative images and time courses of hSyn-RCaMP1 fluorescence would be very helpful.

For instance, panel A mentions “spontaneous neurons and dendrites” but it is hard to visually discriminate any dendrites in the example of hSyn-RCaMP1 fluorescent showed in the Fig.3C. So, showing high-scale representative images for Grn1 KD and control is needed. Similarly, the panels B and C mention the amplitude of spontaneous and stimulation evoked ROI signalling, whereas panel C shows the numbers of responsive vs. non-responsive neurons. These pooled data should be accompanied by the representative examples of base-line and stimulus evoked fluorescent signals both for responsive and non-responsive ROIs. Showing the response heatmaps for the KD and control neurons (as it was done for the astrocytes in the Fig.3) would also be beneficial..

We have added several new panels of example data to Figure 4 that illustrate differences between control and Grin1 KD. Our analysis identifies active areas within the neuronal Ca^{2+} field of view, which often localize to somata, but also occur within the neuropil. We have now revised our use of the term “dendrite” because our imaging was not at sufficient resolution to capture individual dendritic events, though much of the neuropil signal likely originates from the neuronal arbour.

The new example data shows how neuronal activity is recruited in controls with whisker stimulation. Grin1 KD neurons have a high level of spontaneous activity within the same neuronal structures, which is illustrated by the heat map from multiple trials. There is also little neuronal recruitment in Grin1 KD with stimulation. Neurons with calcium events outside the stimulus period (grey shaded area) in Fig. 4F, G are non-responsive cells.

Please also see the example calcium movies of neuronal activity in control (Supplementary Movie 1) and Grin1 KD mice (Supplementary Movie 2).

5) In many figures, the statistical data are presented as violin plots and the statistical significance is said to be calculated using linear mixed models method. Although this method is suitable to verify the statistical difference between KD and control in the context of the work, a wider audience might not be familiar with details of this method, so more detailed explanation of linear mixed models is in order. Also, some explanation of the meaning of X and Y parameters in violin plot would be helpful, either in the legend to some figure (e.g. Fig2) or in Methods.

We have added more information about the design of our linear mixed models and why we used this statistical method in the Material and Methods section. The nested design of our study warranted the use of these tests because our individual data points (n) were not independent (multiple Ca²⁺ events were detected from the same field of view, multiple fields of view were found in each animal, etc.).

We have also updated the figure legends to provide more information about the graph parameters, specifically regarding the data represented in the violin plots and bar graphs.

Reviewer #2 (Remarks to the Author):

In their manuscript "Cortical astrocyte N-Methyl-D-Aspartate receptors influence whisker barrel activity and sensory discrimination" Ahmadpour et al. describe the physiological and behavioral effects of knocking down NMDAR expression in astrocytes, achieved by shRNA hairpins for knockdown of Grin1 mRNA (Grin1 KD). The construct, that also have Lck-GCaMP is delivered by AAV9, reduces the expression of Grin1 in astrocytes by 70%, and decreases NMDAR activity in acute adult brain slices. They further show Grin1KD diminished sensory-evoked calcium events in vivo in astrocytes in the barrel cortex, and different neuronal responses (diminished responses but with higher amplitude) to stimulation in vivo. Then they tested the levels of synchronicity during stimulation: they found that controls had lower BL synchronicity which increased during stimulation, and Grin1KD had higher synchronicity without stimulation, which decreased during stimulation. They then analyze the delayed response from the time of stimulation to the beginning of the Ca²⁺ response, which was the same in neurons, and reduced in astrocytes, injected with the KD virus. Finally, they checked behavior and found that Grin1KD have a problem with detecting small (but not large) grit differences.

I am not impressed by the study, as it has a very preliminary feel. The impression is of a list, rather than a layered work. In the bottom line there is a single 2-photon study (even if it is given in 4 figures), and an additional small behavioral study. This work will require more experiments, and better analyses of the existing one, before I can recommend publication.

Major:

1) I would like to see imaging and behavior joined - there are a lot of studies with head fixed behavior under a 2-photon in the barrel Cx. And then you can really say something about the connection between them.

There are paradigms where animals are trained for a sensory discrimination task during head fixed two photon imaging (for example the sandpaper texture discrimination utilized by Chen et al. ⁵, which is most similar to the behaviour test used in our study). However, it is not clear what new information would be added to this manuscript by combining these techniques. We show that astrocyte Grin1 KD decreases the synchronization of high responding neurons that encode the whisker stimulus, while also decreasing the recruitment of low amplitude neurons. From a behavioural standpoint, this has an impact on sensory acuity because mice can not discriminate between similar textures. In paradigms that have used sandpaper discrimination during imaging, it was found that similar sandpaper textures evoked a comparable calcium amplitude in neurons in animals discerning novel textures ⁵. Therefore, performing a

textural discrimination task during two photon imaging would not add more information than we have already described here about the impairments in neuronal responses to a sensory input after astrocyte Grin1 KD. Future studies that co-label neurons that project to other brain regions (such as S2 and M1) and combine 2-photon with discrimination behaviour could provide insight into how astrocyte NMDA receptors influence specific neuronal populations that have distinct roles in sensory information processing, but this is beyond the scope of this initial study.

2) In addition, I would have liked to see at least a beginning of a mechanism. How do the astrocytes affect neurons? Why in some cases there's a big difference in the astro, but not in the neurons (Fig 6)? Etc...

We are pleased to present new evidence of the mechanism of how astrocytes may modulate neuron activity in this case. In Figure 8, we demonstrate that ATPyS can rescue the neuronal phenotype *in vivo* and improve animal discrimination in the whisker-mediated task (Fig. 8). This suggests that purinergic signaling is lost after astrocyte Grin1 knockdown and future studies will determine the synaptic role of this pathway at specific neuronal connections. Furthermore, these findings are specific to ionotropic mechanisms because disruption of IP3-mediated signaling and reduced astrocyte calcium signaling, does not cause the same neuronal phenotype as NMDA receptor knockdown (Supplementary Fig. 5).

3) You analyze the astro and neurons separately. Why? Is there nothing to be extracted from the fact that you do double imaging?

The purpose of dual imaging in this study was to investigate the impact of reduced astrocyte NMDAR signaling on both astrocytes and neurons in the same animal. Astrocytes and neurons were analyzed separately because they have very different calcium signaling characteristics in terms of their amplitude, duration, and onset relative to the whisker stimulus. We did make temporal comparisons between astrocytes and neurons in the same field of view when considering calcium signal onset time (Fig. 6) because these types of temporal comparisons have been made previously. Further analysis of other parameters such as the spatial correlation between astrocyte and neuronal calcium events do not seem relevant in this case because we see a stark reduction in the number of astrocyte microdomain events and the number of responding neurons after astrocyte Grin1 KD, making it difficult to make spatial comparisons between both cell types.

4) The effect on neurons amplitude (Fig 4) is very small. $P < 0.01$ for more than thousand ROIs per group, is not that impressive (Fig 4D).

Then you divide it by responsiveness, and find changes in ROIs/FOV only in low responsive cells. I understand that the authors give an explanation " Low and mid-responsive neurons are more weakly recruited during repeated trials of sensory stimulation, but they have a greater capacity for reorganization during changes in sensory input". But can they preclude the alternative explanation that this group is comprised from most of the neurons (88%) so it is easier to find significance, but not in the smaller groups?

We agree that the effect on mean neuronal amplitude is very small when you consider all of the neurons together (~1800 control neurons and 1200 Grin1 KD neurons; Fig. 4K), but when we divide the population by responsiveness, there is no change in the amplitudes between groups (low, mid, high responsive and

control vs. Grin1 KD; Fig. 4M). We also considered the number of each of these neuron types per field of view (normalized to the barrel area to account for different field sizes; Fig. 4N). It should be noted that Grin1 KD neurons were classified based on their mean amplitude compared to neurons in the control population (e.g. 88%). It stands to reason that the increased neuronal amplitude in Grin1 KD during whisker stimulation (Fig. 4K) occurs because there is a smaller proportion of low responsive cells compared to mid and high responsive cells. Even though high responsive cells are a much smaller proportion of the population, it is possible to detect differences in this group. In our new data with ATPyS in Fig. 8, we detected an increase in the number of low responsive cells in Grin1 KD and decreases in mid and high responsive cells.

5) It is very difficult to understand the figures.

For example Fig 3D, Fig4B,D – what's in the grey part? Is it all the data (i.e. ROIs) or the data per mouse? Fig 4 (as another example): Dark purple color is used to mark 2 different things (F and the rest), and Grin1KD is light purple, except than in E.

We have provided further clarification about the figures in the figure legend (e.g. the grey part is the mean of all data (not per mouse)) and updated the use of colour for clearer representation.

6) Fig 6 – if the astrocyte are changed, with no effect on the neurons – what does it mean??

We did not observe a difference in onset time of neurons that responded to whisker stimulation in Grin1KD vs. controls (Fig. 6B). This suggests that in the remaining population of responding neurons that can still respond to whisker stimulation (likely high responders, Fig. 4), there is no delay in their action potentials. This onset timing of neurons was used to define the onset of different astrocyte populations (fast and delayed; Fig. 6C). We observed a reduction in the number of astrocyte calcium events in both of these groups.

To put it simply, a reduction in astrocyte MDs, without an effect on neuronal onset times, means that astrocytes do not affect the temporal onset of neurons that respond to stimulation.

However, we show elsewhere in the manuscript that astrocytes can affect neuronal activity during stimulation in other ways (recruitment, adaptation, synchronization). Rapid onset astrocyte microdomains as described in Fig. 6, would have the necessary temporal dynamics for rapid gliotransmission and neuromodulation. Perhaps a loss of these microdomains reduces purinergic gliotransmission altering the neuronal circuit as described above.

Minor

1) The Abstract doesn't do the paper a good service...

We have now revised the abstract.

2) Figure 1 – it's really not necessary to show the controls in the main figure...

The control pictures have been moved to Supplementary Figure. 1

3) P 8, row 6 "...99th" do you mean 88th?

Yes, this was an error. It is now fixed.

4) The discussion is VERY long, and have little that was not said before.

We have revised the discussion to make it more succinct.

Reviewer #3 (Remarks to the Author):

The functional characterization of neuronal NMDA receptors has been awarded a Nobel Prize and has become textbook knowledge for biology students. Much less is known about NMDA receptors in glial cells. The work described in the manuscript by Noushin Ahmadpour and colleagues significantly adds to our current understanding of NMDA receptors in a major subtype of glial cells, astrocytes. The authors used a multi-shRNA AAV approach to knockdown the constitutive NMDA receptor subunit GriN1 in astrocytes and then employed a combination of whisker stimulation and in vivo two-photon calcium imaging to investigate the impact of NMDA receptor-mediated astroglial calcium signals, neuronal activity, and somatosensory behavior.

The main observations are:

1. AAV-shRNA reduces GriN1 mRNA levels by about 70% in FAC-sorted astrocytes.
2. In brain slices, fewer NMDA/glycine-evoked calcium signals are found compared to control preparations.
3. In awake animals, whisker stimulation cannot trigger higher amplitude calcium signals in GriN1 knockdown astrocytes.
4. The Ca²⁺ signals of neurons with high responsiveness to whisker stimulation are blocked in astroglial GriN1 knockdown barrel fields.
5. Whisker stimulation in control mice evoked more synchronous calcium signals in neurons than in those of astroglia-GriN1 deficient mice.
6. The numbers of stimulus-evoked calcium microdomains were reduced in GriN1-knockdown astrocytes versus controls.
7. In a somatosensory behavioral experiment (detection of sandpaper with differently sized grains by the whiskers) the mice with GriN1 deficiency were not able to distinguish the sandpapers. However, the sandpaper itself elicited neuronal and astroglial calcium responses.

From their observations the authors conclude that astrocytes detect distinct neuronal activity via their NMDA receptors. The astrocytes then enhance synchronization of adjacent neurons. The combined neuronal and astroglial communication is essential to distinguish fine differences in whisker stimulation.

This is a very interesting observation and, so far, the best experimental evidence how astroglial NMDA receptors contribute to a distinct brain function.

In general, the work is well described and performed, although some critical points remain:

1. The extent of astroglial GriN1 knockdown has only been shown by a qPCR of GriN1 from FAC sorted astrocytes. The functional analysis only indirectly suggests that also the receptor channel complex is missing. Here, at least two of three types of additional experiments are required: 1. Whole-cell patch-clamp recordings of control and mutant astrocytes in slices to complement the calcium traces of fig. 2 C/D. In addition, the time scale should be magnified. 2. Immunohistochemistry of GriN1 on control and mutant cells. The cell-specific localization of GriN1 on astrocytes will be facilitated by the membranous Lck-GCaMP expression. 3. qPCR of the NMDA receptor subunits to check for compensatory mechanisms. These additional experiments are necessary since about 50 % mutant and GriN1-KD astrocytes display the same level of GriN1 mRNA. The patch-clamp and/or the immunohistochemical analysis would indicate to which extent the functional receptor is degraded and removed from the membrane.

We now provide new data that addresses following:

- 1) We transduced astrocyte-enriched glial cultures with our control and Grin1 KD viruses and stained for GluN1 using immunocytochemistry. Cellular intensity quantification determined there was less GluN1 staining in Grin1 KD astrocytes (expressing the virus and colocalized with GFAP; Fig. 1H-J).
- 2) We made multiple attempts with different antibodies to stain astrocyte GluN1 in fixed cortical tissues using the Lck-GCaMP6f expression to identify membranous astrocytes areas, as suggested by the Reviewer. This proved to be a very challenging experiment because GluN1 signal contamination from the “neuropil” was too high and made it impossible to discern a difference between control and Grin1 KD astrocytes by conventional confocal microscopy. Therefore, we took a different approach and depleted neuronal GluN1 within our AAV virus injection area. This was achieved by mixing a neuronal Cre virus (AAV-hSYN-Cre-mCherry) with our astrocyte control or Grin1 KD virus and injecting this mix into the cortex of mice with floxed Grin1 and an EYFP reporter. The EYFP expression activated by the neuronal Cre virus helped us to identify the Cre positive neurons and comparing to the GluN1 intensity outside the virus area, we confirmed that there was reduced GluN1 staining within the neuropil in the virus area. Subsequently, using GFAP to create a mask of individual cortical astrocytes in L2/3, we found a decrease in GluN1 fluorescence in Grin1 KD astrocytes (Fig. 1K-L).
- 3) We did RNA sequencing with the RNA samples collected from FACS sorted astrocytes and we compared the expression levels of the other NMDA receptor subunits. There was no significant increase in any subunits that would suggest compensation, but Grin2a was also reduced in Grin1 KD samples (Fig. 1F, G).

It should also be noted that new calcium traces are provided in Figure 2 with an expanded time scale as mentioned above.

2. A more detailed localization of GriN1 should be deduced from a cellular map of microdomains along processes and at process tips, in particular to distinguish perisynaptic from endfeet signals. For that purpose, from control and mutant astrocytes the individual calcium microdomains should be detected and, for example, plotted as a localization heatmap onto the outline of schematic astrocyte.

In order to properly localize the Lck-GCaMP6f activity to different astrocyte compartments, we labelled astrocytes with SR101. This permitted clearer identification of somata and endfeet. New panels in Figure 3 provide example traces from each of these compartments and quantification of how the level of calcium activity changes in these regions after Grin1 KD.

3. Why were some complete sets of experiments (slice imaging, whisker discrimination) only performed with female mice?

With the data added to the paper during revision, males and females are now included in all experiments, apart from the GluN1 immunohistochemistry, IP3R2 KO and qPCR/RNA work. For these experiments all animals were female. This was partly because these were the animals that we had on hand at the time of the experiments (particularly mice from the breeding colonies for floxed NR1 mice and IP3R2 KO), but we were concerned that mixing sexes would add to the variability of our results when the FACS sorting/ RNA work was already inherently variable from multiple handling steps. We have made efforts to clearly indicate the sexes of animals used in all experiments in Table 1 and provide data in Supplementary Figure 4 that shows no sex effects for the astrocyte and neuron calcium events.

Minor points:

Figure 6: The title of the legend is misleading. All type of microdomains are reduced, not only those with rapid onset. In A, the description of the X-axis is missing.

We agree with the reviewer and we have updated the figure legend to better reflect the data.

References

1. Stobart, J. L. *et al.* Cortical circuit activity evokes rapid astrocyte calcium signals on a similar timescale to neurons. *Neuron* **98**, 726-735.e4 (2018).
2. Stobart, J. L. *et al.* Long-term in vivo calcium imaging of astrocytes reveals distinct cellular compartment responses to sensory stimulation. *Cereb. Cortex* **28**, 184–198 (2018).
3. Srinivasan, R. *et al.* Ca²⁺ signaling in astrocytes from Ip3r2 ^{-/-} mice in brain slices and during startle responses in vivo. *Nat. Neurosci.* **18**, 708–717 (2015).
4. Agarwal, A. *et al.* Transient opening of the mitochondrial permeability transition pore induces microdomain calcium transients in astrocyte processes. *Neuron* **93**, 587-605.e7 (2017).
5. Chen, J. L. *et al.* Pathway-specific reorganization of projection neurons in somatosensory cortex during learning. *Nat. Neurosci.* **18**, 1101–1108 (2015).

REVIEWERS' COMMENTS

Reviewer #1 (Remarks to the Author):

After the revision, the paper has undergone significant changes and its technical quality and overall message has substantially increased. Authors have performed a great deal of work to address the reviewers' comments and now present several new lines of evidence which make conclusions of the work really sound and coherent.

In particular, new lines of experiments comparing Grin1 KD mice with mice lacking astroglial IP3R2 receptors allow to discriminate contribution of inotropic (NMDAR-mediated) vs metabotropic signalling into astroglial function and glia-neuron interactions. The data on essential contribution of glial NMDARs, provided in the paper, can also help to resolve long-standing debate on physiological importance of astrocytic calcium signalling which were triggered by smaller-than-expected effects of astroglia-specific IP3R2 knockout. Another new and impressive result is a clear demonstration that effects of astroglial NMDARs on neuronal function are mediated via release of ATP from astrocytes.

These results significantly increase the importance of this work for the research filed.

Overall, the revised manuscript gains significantly on the quality and importance and makes substantial advance in the field.

Reviewer #2 (Remarks to the Author):

The revised manuscript is significantly improved (especially the new fig. 8).

The questions were partially answered, but I accept the authors claim that some of them may be outside the scope of this paper. Therefore, I will be happy to see it published in Nature Communications.

Reviewer #3 (Remarks to the Author):

The critical points of this reviewer have been addressed.

This new work of the Stobart lab further confirms the crucial role of astrocytes in modulating neuronal activity and synchronization. Importantly, the authors assigned a distinct in vivo function to astroglial expression of the Grin1 gene. Knockdown of Grin1 in astrocytes altered neuronal function, as evidenced by the functional reduction in astrocyte Ca²⁺ responses to NMDAR agonists and slower adaptation of neuronal Ca²⁺ events in Grin1 KD mice during prolonged electrical stimulation of the whisker pad. Astrocyte Grin1 KD also alters neuronal synchronization and adaptation, with high-responsive neurons being less synchronized during whisker stimulation. These important findings significantly contribute to a better understanding of the complex interplay between astrocytes and neurons, although molecular details linking astroglial NR1 signaling and ATP release still have to be addressed.

Response to Referees

Reviewer #1 (Remarks to the Author):

After the revision, the paper has undergone significant changes and its technical quality and overall message has substantially increased. Authors have performed a great deal of work to address the reviewers' comments and now present several new lines of evidence which make conclusions of the work really sound and coherent.

In particular, new lines of experiments comparing Grin1 KD mice with mice lacking astroglial IP3R2 receptors allow to discriminate contribution of ionotropic (NMDAR-mediated) vs metabotropic signalling into astroglial function and glia-neuron interactions. The data on essential contribution of glial NMDARs, provided in the paper, can also help to resolve long-standing debate on physiological importance of astrocytic calcium signalling which were triggered by smaller-than-expected effects of astroglia-specific IP3R2 knockout. Another new and impressive result is a clear demonstration that effects of astroglial NMDARs on neuronal function are mediated via release of ATP from astrocytes. These results significantly increase the importance of this work for the research field.

Overall, the revised manuscript gains significantly on the quality and importance and makes substantial advance in the field.

We thank Reviewer 1 for their positive comments.

Reviewer #2 (Remarks to the Author):

The revised manuscript is significantly improved (especially the new fig. 8).

The questions were partially answered, but I accept the authors claim that some of them may be outside the scope of this paper. Therefore, I will be happy to see it published in Nature Communications.

We thank Reviewer 2 for their understanding regarding our previous revisions and their encouraging comments.

Reviewer #3 (Remarks to the Author):

The critical points of this reviewer have been addressed.

This new work of the Stobart lab further confirms the crucial role of astrocytes in modulating neuronal activity and synchronization. Importantly, the authors assigned a distinct in vivo function to astroglial expression of the Grin1 gene. Knockdown of Grin1 in astrocytes altered neuronal function, as evidenced by the functional reduction in astrocyte Ca²⁺ responses to NMDAR agonists and slower adaptation of neuronal Ca²⁺ events in Grin1 KD mice during prolonged electrical stimulation of the whisker pad.

Astrocyte Grin1 KD also alters neuronal synchronization and adaptation, with high-responsive neurons being less synchronized during whisker stimulation. These important findings significantly contribute to a better understanding of the complex interplay between astrocytes and neurons, although molecular details linking astroglial NR1 signaling and ATP release still have to be addressed.

We thank Reviewer 3 for their support and we agree that the molecular link between astrocyte NMDA receptors and extracellular ATP signaling will be an important follow-up study.